# An FPGA-based analysis of trade-offs in the presence of ill-conditioning and different precision levels in computations

Ignacio Algredo-Badillo[1], José Julio Conde-Mones[2]*, Carlos Arturo Hernández-Gracidas[3]*, María Monserrat Morín-Castillo[4]*, José Jacobo Oliveros-Oliveros[2], Claudia Feregrino-Uribe[5]

**1** Computer Science Department, CONACYT-INAOE, Tonantzintla, Puebla, México, **2** Physical-Mathematical Science Faculty, BUAP, Puebla, Puebla, México, **3** Physical-Mathematical Science Faculty, CONACYT-BUAP, Puebla, Puebla, México, **4** Electronic Science Faculty, BUAP, Puebla, Puebla, México, **5** Computer Science Department, INAOE, Tonantzintla, Puebla, México

* juliocondem@hotmail.com (JJCM); carloshg@fcfm.buap.mx (CAHG); maria.morin@correo.buap.mx (MMMC)

**Data Availability Statement:** All relevant data are within the paper and its Supporting Information files.

## Abstract

Several areas, such as physical and health sciences, require the use of matrices as fundamental tools for solving various problems. Matrices are used in real-life contexts, such as control, automation, and optimization, wherein results are expected to improve with increase of computational precision. However, special attention should be paid to ill-conditioned matrices, which can produce unstable systems; an inadequate handling of precision might worsen results since the solution found for data with errors might be too far from the one for data without errors besides increasing other costs in hardware resources and critical paths. In this paper, we make a wake-up call, using 2 × 2 matrices to show how ill-conditioning and precision can affect system design (resources, cost, etc.). We first demonstrate some examples of real-life problems where ill-conditioning is present in matrices obtained from the discretization of the operational equations (ill-posed in the sense of Hadamard) that model these problems. If these matrices are not handled appropriately (i.e., if ill-conditioning is not considered), large errors can result in the computed solutions to the systems of equations in the presence of errors. Furthermore, we illustrate the generated effect in the calculation of the inverse of an ill-conditioned matrix when its elements are approximated by truncation. We present two case studies to illustrate the effects on calculation errors caused by increasing or reducing precision to *s* digits. To illustrate the costs, we implemented the adjoint matrix inversion algorithm on different field-programmable gate arrays (FPGAs), namely, *Spartan-7*, *Artix-7*, *Kintex-7*, and *Virtex-7*, using the full-unrolling hardware technique. The implemented architecture is useful for analyzing trade-offs when precision is increased; this also helps analyze performance, efficiency, and energy consumption. By means of a detailed description of the trade-offs among these metrics, concerning precision and ill-conditioning, we conclude that the need for resources seems to grow not linearly when precision is increased. We also conclude that, if error is to be reduced below a certain threshold, it is necessary to determine an optimal precision point. Otherwise, the system becomes more sensitive to measurement errors and a better alternative would be to choose precision

**Funding:** J.J.O. received financial support from the Meritorious Autonomous University of Puebla (BUAP), through project VIEP-00174 (Web page: https://www.buap.mx/). I.A. and C.A.H. are commissioned to their respective institutions through the National Council of Science and Technology (CONACYT) Research Fellows program, by means of projects 882 and 278, respectively (Web page: https://www.conacyt.gob.mx/). The funders had no role in study design, data collection and analysis, decision to publish, or preparation of the manuscript.

**Competing interests:** The authors have declared that no competing interests exist.

carefully, and/or to apply regularization or preconditioning methods, which would also reduce the resources required.

## Introduction

Throughout the history of modern society, we can find several examples of critical failures, which are attributed to errors in calculations due to limited resources.

One representative example of this is the case of the Patriot missile failure, which occurred in 1991 during the Gulf War when a Patriot Missile launched by the US Army failed in its purpose of intercepting a Scud missile, launched by the Iraqi army. The Scud missile hit an Army barrack in Saudi Arabia territory and had the consequence of 28 soldiers killed and a much bigger number of injured people. The failure, according to the official report [1], was due to a software problem, caused by an inaccurate calculation of time, which was exacerbated by the fact that the missile battery had been continuously operating for over 100 hours. The accumulated error in the calculation caused the launched Patriot Missile to miss the Scud it was aimed to. A reboot every few hours was needed to alleviate the problem.

Surprisingly, instances wherein different types of digital devices run out of resources or miss calculations are still frequent. However, in several of these cases, the problems are caused by inadequate handling of ill-conditioning which can produce large errors in the solution to the systems of algebraic equations when there are errors in the input data, rather than by an inappropriate solution per se.

Matrices are an important tool for solving various mathematical problems. Some types of problems are modeled directly by systems of linear algebraic equations, which can be large, whereas some systems of algebraic equations have (ill-conditioned) matrices that can be obtained when some linear operational equations in infinite dimensions are discretized. These operational equations can be ill-posed in the sense of Hadamard [2], which leads to ill-conditioned matrices, and can be derived when a problem is modeled by integral or differential equations. An example in this regard is inverse electroencephalography (inverse EEG), which is presented in the following section. Electrical activity is generated by the bioelectrical activity of a large population of neurons working synchronously [3, 4], which is recorded by electrodes located on the scalp using electroencephalography (EEG). EEG is related to the bioelectrical sources using a model that considers the head as a conductive inhomogeneous medium of multiple layers that represent the different regions of the head, i.e., brain, skull, and scalp. This model, and the quasi-static approximation of the Maxwell equations [3, 5], lead to a boundary value problem defined in an inhomogeneous medium with appropriate boundary conditions. It is known that to define an operator we need a domain, a codomain, and a correspondence rule. In the case of EEG, the operator is defined in the following way:

- The operator has as domain and codomain appropriate Hilbert spaces [5, 6].

- The correspondence rule is given by: this operator associates the bioelectrical source with the trace of the solution to the boundary value problem (the restriction to the boundary of the region that represents the head).

Another example is given in electrocardiography (ECG); specifically, inverse ECG consists of reconstructing cardiac electrical activity from given body surface electrocardiographic measurements (see Fig 1(a)). These measurements are related to the potential using a boundary value problem defined in an annular region. This region is related to the Cauchy problem for

(a) (b)

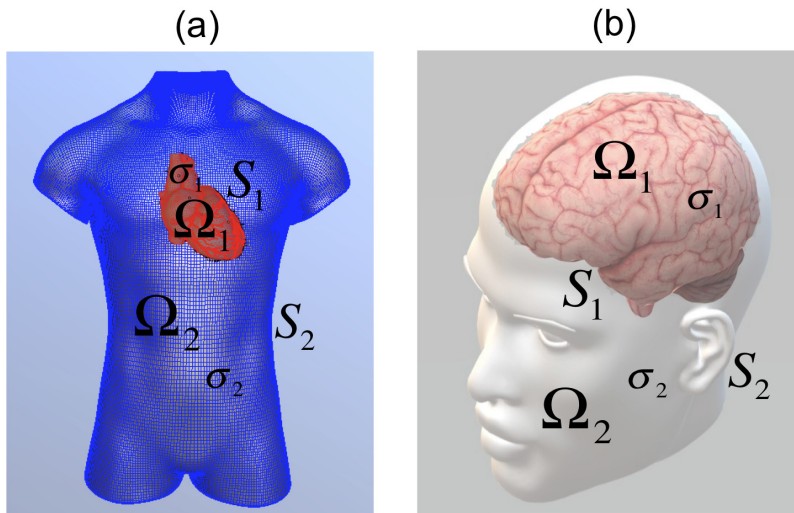

**Fig 1. Region** $\Omega = \Omega_1 \cup S_1 \cup \Omega_2$. The source $g$ is defined on the boundary $S_1$ for (a) ECG and (b) EEG.

the Laplace equation using a boundary value problem, which allows an operational statement of the form $K\varphi = V$ to be made, where $K$ is a linear, injective, and compact operator, while $\varphi$ and $V$ belong to appropriate Hilbert spaces [7]. This operational equation, which is ill-posed in the sense of Hadamard [2], can be discretized, leading to a linear system of equations with an ill-conditioned matrix that can result in numerical instability in the presence of errors. Notably, the operator $K$ is linear, compact, and injective, and $\varphi$ and $V$ belong to appropriate Hilbert spaces of infinite dimension, which imply that $K^{-1}$ (the inverse operator of $K$) is not continuous [2]. Therefore, the matrices obtained when the operational equation $K\varphi = V$ is discretized are ill-conditioned. These operational equations appear in various practical problems, such as the source and potential identification problems. To handle this numerical instability, regularization methods can be employed, such as Tikhonov, Lavrentiev, and Landweber. These methods have been applied to several practical problems, e.g., inverse EEG, ECG, problems in geophysics, the Cauchy problem, and the Laplace and Helmholtz equations among others [2, 3, 7–12]. Furthermore, these methods allow the numerical instability present in the ill-posed problems (in the sense of Hadamard) [2] to be addressed. Other applications include photoacoustic tomography and tomography imaging. In the former case, the Tikhonov filter uses a temporal data deconvolution method in the filtered back-projection algorithm [13]. In the latter, due to the physical conditions of the data acquisition process, it is common to find a noisy, incomplete set of unequally spaced projections wherein this problem is ill-posed [14]. One more application in which matrices are used consists of Polynomial probability distribution estimation based on $N$ statistical moments from each distribution, which is essential in applied statistical analysis in diverse scientific fields [15].

In this work, several analyses of ill-conditioned matrix inversion are presented in terms of trade-offs among precision, throughput, efficiency, error, and area, where diverse field-programmable gate array (FPGA) technologies are evaluated. Specifically, we emphasize that two important problems may arise when dealing with precision in computations:

- There are several cases (ill-conditioned matrices) where the results will not improve, no matter how much precision is increased, and increasing precision will make a system more sensitive to measurement or input data errors (this is shown in Fig 2). Furthermore, if the

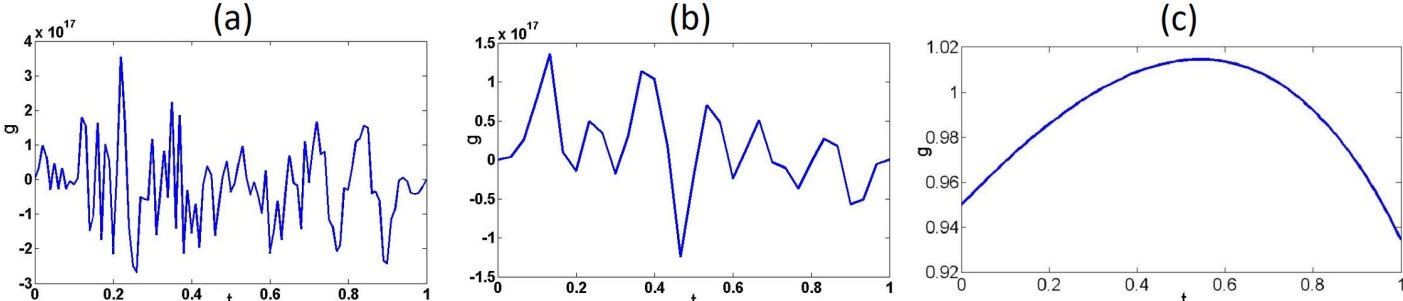

**Fig 2. Solutions to the system in Eq (8).** Approximate solutions without regularization for (a) $n = 100$ and (b) $n = 30$. (c) Approximate solution with regularization for $n = 30$.

elements of the matrix are approximated to $s$ digits using inadequate precision the computed solution to the disturbed system can be too far from the exact solution to the undisturbed system, as shown in Table 3.

- We must consider cases where the solution might require increasing precision to an adequate number of $s$ digits, where embedded systems and mobile devices might have insufficient available hardware resources, which becomes an infeasible option (as exemplified in Table 2). In these cases, a limited number of input and output blocks (IOBs) could be used for implementation.

These problems are the motivation for this study, given the relevance of analyzing the behavior of a system when precision is increased, along with the different trade-offs related to this increment, especially when effects such as ill-conditioning cause a system to deliver inadequate results. The rest of the work presented in this article is devoted to answering the following research question:

- How does the presence of ill-conditioning affect the optimal number of decimal digits used when calculating an inverse matrix, and what would be the consequences, in terms of resource consumption, if precision in such calculations was increased?

With respect to the first part of the question, we developed ad hoc examples, covering two representative case studies for extreme situations: one where truncation is performed with too few decimal digits, and another where an adequate number of decimal digits is used. These examples are intended to connect ill-conditioning and precision with consumption of hardware resources. Regarding the second part of the question, different implementations using several different FPGA technologies are presented to evaluate the impact of the use of different precision levels. We compared the main characteristics of our implementations, such as matrix size, solution algorithm, and resource consumption with respect to similar state-of-the-art solutions.

The main contributions of the results in this study are as follows:

- Two case studies showing an analysis of trade-offs in varying truncation and precision on calculations of the inverse matrix and the solution to the associated system of equations are presented. In the first case, using too few decimal digits causes error amplification; in the second case, an adequate number of decimal digits produces a more accurate solution.

- The adjoint matrix inversion method was implemented for a $2 \times 2$ matrix, where different levels of precision and devices were used, which allowed us to thoroughly analyze trade-offs

at different precision levels. With this, we were able to measure how changing precision affects the resources needed for calculations, wherein limited resources might constrain implementation of certain algorithms.

- The most important contribution of this work consists of connecting conclusions from the ill-conditioning case studies with those derived from hardware implementations. Moreover, we can consider the connection is between real-life problem-solving and hardware resources. In this work, the effects of precision in the calculations in terms of resource consumption on the algorithms have been clarified through implementation on different hardware technologies; particularly, the demonstration with the calculation of the inverse of a $2 \times 2$ matrix using the adjoint matrix inversion method. These effects can considerably be aggravated with erroneous calculation results when the matrix is ill-conditioned and this is not taken into account in the implementation of the algorithm.

These results, although determined from specific case studies, can be easily extrapolated to real-life problems.

In the following section, we explain the primary concepts of operational equations of the first kind. We then proceed to explain the relevance of precision in digital calculations, followed by the methods used for the analysis performed. We then present our experimental results, which is followed by a discussion section. Finally, we present the conclusions drawn from this work.

## Operational equations of the first kind and their discretization

Integral equations of the form

$$\int_a^b k(t,s)f(s)ds = g(t) \tag{1}$$

appear in several applications, where $t \in [c, d]$ and $k(t, s)$ is called the kernel of the operator. Some examples are: a) $k(t, s) = \frac{1}{|t-s|}$, which corresponds to the case of an electric potential, and b) $k(t, s) = \exp\left(-\frac{(t-s)^2}{2\beta^2}\right)$, which is a kernel that models (in one dimension) the long-time average effects of atmospheric turbulence on light propagation [16]. In general, Eq (1) can be written as follows:

$$\mathbf{K}(f)(t) = g(t) \tag{2}$$

where the operator $\mathbf{K}$ acts between appropriate spaces. The operational form of Eq (2) is used to study inverse EEG and ECG, in optics, in inverse geophysical and atmospheric studies, and in other applications [3, 10–12, 16, 17]. As an example of this, the potential produced in the head by a bioelectrical source $g$ located on the cerebral cortex of the brain is given by the following boundary value problem: Find $u_1$ and $u_2$ such that

$$-\sigma_1 \Delta u_1 = 0 \qquad \text{in} \quad \Omega_1, \tag{3}$$

$$-\sigma_2 \Delta u_2 = 0 \qquad \text{in} \quad \Omega_2, \tag{4}$$

$$u_1 = u_2 \qquad \text{on} \quad S_1, \tag{5}$$

$$\sigma_1 \frac{\partial u_1}{\partial \mathbf{n}_1} = \sigma_2 \frac{\partial u_2}{\partial \mathbf{n}_2} + g \qquad \text{on} \quad S_1, \tag{6}$$

$$\sigma_2 \frac{\partial u_2}{\partial \mathbf{n}_2} = 0 \qquad \text{on} \quad S_2, \tag{7}$$

where $\Omega_1 \cup S_1 \cup \Omega_2 \subset \Re^d$, with $d = 2$ or 3, is a sufficiently smooth region. The positive constants $\sigma_1$ and $\sigma_2$ are the conductivities of regions $\Omega_1$ and $\Omega_2$, respectively (see Fig 1(b)). The function $g$ represents the source defined on the interface $S_1$, while $\mathbf{n}_1$ and $\mathbf{n}_2$ are the unit outward normal vectors on the boundary of $\Omega_1$ and $\Omega_2$, respectively. Note that, in particular, $\mathbf{n}_2 = -\mathbf{n}_1$ on the interface $S_1$. Let $u$ be the solution to problem (3)–(7) in $\Omega$ and define $u_i = u|_{\Omega_i}$, $i = 1, 2$. $\Delta$ represents the Laplace operator, which is also denoted by $\nabla^2$. An operator $A$ is defined by the following correspondence rule: this operator associates the bioelectrical source with the trace (the restriction to the boundary of $\Omega$) of the solution to problem (3)–(7). The operator has as domain and codomain appropriate Hilbert spaces [5, 6]. Using the previous correspondence rule, it is possible to define an operational equation, that has the form of (2), and that allows the problem of identifying sources defined on $S_1$ to be studied. The inverse operator $A^{-1}$ is not continuous and leads to numerical instability in the sense of Hadamard of the problem, which is the cause of ill-posedness [2].

Eq (2) can be discretized in the following form:

$$Kx = y \tag{8}$$

where $K$ is a matrix such that $K_{ij} = k(t_i, s_j)$, $\{t_i\}_{i=1}^n$ and $\{s_j\}_{i=1}^n$ are partitions of the intervals [$a$, $b$] and [$c$, $d$], $x = (f(s_1), f(s_2), \ldots, f(s_n)))$, and $y = (g(t_1), g(t_2), \ldots, g(t_n)))$, respectively. To determine an approximate solution to the operational equation, we must find the solution to the corresponding system of algebraic equations. As an example, we consider the following integral operator $\mathbf{K}$: $L_2(0, 1) \to L_2(0, 1)$, such that

$$\mathbf{K}(f)(t) = g(t) = \int_0^1 (1 + ts)e^{ts}f(s)ds, \quad 0 < t < 1 \tag{9}$$

where $L_2(0, 1)$ represents the space of square functions. The operator $\mathbf{K}$ is injective [2]. We consider a regular partition of the interval [0, 1] of length $h$ for both $s$ and $t$, i.e., the same partition is used for the integration interval and the evaluation of function $g$. The matrix of the system is given by $K_{ij} = h(1 + t_i s_j)e^{t_i s_j}$. The solution to the integral equation in (9) is $f(s) = 1$ for $g(t) = e^t$. Fig 2(a) shows a plot of the solution to (8) for $n = 100$, and Fig 2(b) shows a plot of the solution for $n = 30$. Similar results are obtained for different values of $n$. Notice that the approximate solution, which was obtained from the corresponding system of algebraic equations, is far from the exact solution ($f(s) = 1$). This numerical instability is a consequence of the high condition number of the matrix corresponding to the system in (8). For $n = 100$, the system was solved using *MATLAB*, where $cond(A) = 7.0253 \times 10^{18}$. For $n = 30$, $cond(A) = 2.3896 \times 10^{18}$ (in both cases, $cond(A)$ was obtained using *MATLAB*'s *cond* function). The integral Eq (9) is ill-posed in the sense of Hadamard because the inverse of the operator is not continuous, which leads to numerical instability when the right side of the equation has errors. Numerical instability can be handled using the Tikhonov regularization method, which consists of minimizing the following functional: $J_\alpha(x) = \|\mathbf{K}x - y\|^2 + \alpha\|x\|^2$, where $\|y - y^\delta\|_{L2(0,1)} < \delta$, and $\alpha > 0$ is called the Tikhonov regularization parameter, which must be chosen in terms of $\delta$. If the regularization parameter is chosen such that $\frac{\delta^2}{\alpha(\delta)} \to 0$ when $\delta \to 0$, then $x^\alpha \to x$, where $x$ is the exact solution for the exact right side $y$ [2]. Fig 2(c) shows the

regularized solution using the Tikhonov regularization method for $\delta = 0.1$ and $\alpha = 10^{-5}$. Vector $y^{\delta}$ was obtained by adding a random vector using *MATLAB*'s *rand* function.

In the following sections, we present some problems that appear in these systems, which are related to their numerical instability and create some difficulties in the hardware implementation, such as, the need to increase precision, and as consequence of this, to require more hardware resources as presented in the discussion section.

## Precision

To understand the relevance of precision, it must be distinguished from accuracy. Accuracy refers to how close a measured value (e.g., an EEG or ECG voltage) is to its true value, while precision refers to repeatability when a value is measured (how close a new measured value will be to a previously obtained value). When measuring accuracy, random and systematic effects cause displacement from the actual value. Precision describes the variability of repeated measurements when the same measurement method is used. Under these conditions, increasing precision is useful for providing consistency when values are repeatedly measured, as in the aforementioned cases of EEG or ECG, where voltage data are obtained over time when measuring the electrical activity of the brain and heart, respectively. Low precision not only reduces the amount of resources needed significantly, but also tends to generate inadequate results in certain applications. It is generally thought that, by increasing precision in computations, results will automatically improve, despite the increase in required hardware resources or energy consumption.

In several applications and digital systems, such as FPGAs and microcomputers, an important numeric representation corresponds to fixed-point notation, where $p$ bits can represent up to $2^{p}$ different consecutive integers. To represent negative numbers, the first bit is used to indicate the sign, and $2^{p-1}$ positive and $2^{p-1}$ negative integers can be represented. If an integer, with a corresponding fraction, is to be represented using fixed-point notation, the $p$ bits are divided into two parts: the one to the left of the implied (fixed) binary point represents the integer part, while the one to its right represents the fraction.

Floating-point notation, on the other hand, is used to represent rational numbers, with a mantissa ($M$) and an exponent ($E$) (in a similar fashion to scientific notation). Using this notation, $M \times 2^{E}$ is a valid representation of a number; and the number of bits $m$ and $e$ (the precision of the representation), reserved for the mantissa and exponent, respectively, determines how many different numbers can be represented. As in the case of fixed-point notation, the first bit ($S$) can be used to indicate the sign of the number. In addition, the exponent $E$ is a signed number where, if $E$ is negative, the represented number is close to 0.

Popular precision levels used for floating point calculations are i) half precision (HP), ii) single precision (SP), iii) double precision (DP), iv) extra precision (EP), and v) quadruple precision (QP). Further details are provided in Table 1.

When comparing fixed-point and floating-point representations, the latter is more exact and allows the representation of bigger and smaller numbers than the former. With the drawback that its software and hardware implementations are more complex, which necessitates the analysis and development of new methods to improve the performance while using floating-point representations.

## Methods

The design exploration carried out in this article focuses on analyzing the trade-offs found when precision should be increased or decreased while calculating the inverse matrix in the

**Table 1. Main characteristics of popular precision levels.**

| Precision | Bits for sign (s) | Bits for exponent (e) | Bits for mantissa (m) | Total bits | Smallest positive number | Largest positive number |
|---|---|---|---|---|---|---|
| Half | 1 | 5 | 10 | 16 | $2^{-14} \approx 6.1 \times 10^{-5}$ | $(2 - 2^{-10}) \times 2^{15} \approx 6.5 \times 10^4$ |
| Single | 1 | 8 | 23 | 32 | $2^{-126} \approx 1.2 \times 10^{-38}$ | $(2 - 2^{-23}) \times 2^{127} \approx 3.4 \times 10^{38}$ |
| Double | 1 | 11 | 52 | 64 | $2^{-1022} \approx 2.2 \times 10^{-308}$ | $(2 - 2^{-52}) \times 2^{1023} \approx 1.8 \times 10^{308}$ |
| Extra | 1 | 15 | 64 | 80 | $2^{-16382} \approx 3.4 \times 10^{-4932}$ | $(2 - 2^{-64}) \times 2^{16383} \approx 1.9 \times 10^{4932}$ |
| Quadruple | 1 | 15 | 112 | 128 | $2^{-16382} \approx 3.4 \times 10^{-4932}$ | $(2 - 2^{-112}) \times 2^{16383} \approx 1.2 \times 10^{4932}$ |

Bits reserved for the sign, exponent, and mantissa for half, single, double, extra, and quadruple precision. The largest and smallest positive numbers that can be represented are also shown for reference.

presence of ill-conditioning. We will observe how truncation affects systems in a different form when it appears in different precision levels.

Precision is key because it directly affects computation rate in processing, control, automation, and optimization problems, among others. In addition, one of the current research directions in embedded systems is the development of applications for Internet of Things (IoT), domotics, Industrial IoT (IIoT), and Industry 4.0 technologies; therefore, depending on the particular problem, each architecture must be designed to meet a set of specifications or requirements.

This emergence of embedded systems and high precision applications is a strong motivation for analysis of two aspects of matrix calculations: i) the trade-offs that exist when precision in ill-conditioned matrices is modified; ii) how available hardware resources, such as memory, number of gates, and power consumption change, and their effects on throughput and efficiency.

For the analysis developed in this article, the adjoint matrix inversion method is implemented for a matrix $A$: $A^{-1} = \dfrac{1}{det(A)} adj(A) = \dfrac{1}{ad - bc} \begin{bmatrix} d & -b \\ -c & a \end{bmatrix}$.

As a matter of fact, there are other alternatives for calculating the inverse of a given matrix, such as Gauss-Jordan elimination and QR decomposition (where a matrix $M$ is decomposed into a product $M = QR$ of an orthogonal matrix $Q$ and an upper triangular matrix $R$). Moreover, some methods are specifically designed for calculating the inverse of ill-conditioned matrices in an approximate way, as shown in [18]. Nevertheless, using any of such methods would not allow assessing the effects of ill-conditioning in results as we intend. In particular, the solutions to the system of linear equations for data with errors and without errors can be far from each other.

The hardware architecture presented for inversion of a $2 \times 2$ matrix uses the full unrolling technique to perform the operations in parallel, as shown in Fig 3. In the diagram, shaded blocks represent hardware processing modules, and yellow blocks are IOBs, whereas white blocks are Simulink modules. The first two block types (shaded and yellow) are modules that constitute the hardware architecture, while the third block type (white blocks) enables signals that are external to the architecture. This last block type is used to represent the functionality of the implemented algorithm.

For simplicity, the inverse of a fixed-size matrix ($2 \times 2$) was chosen as an example, and the adjoint matrix inversion method was used. Our main interest is centered in two aspects: i) determining the effects of truncation error; ii) how resource requirements grow during computation of the inverse matrix as precision increases. How requirements grow with the matrix size was not considered. The selected matrix size and the implemented adjoint matrix method

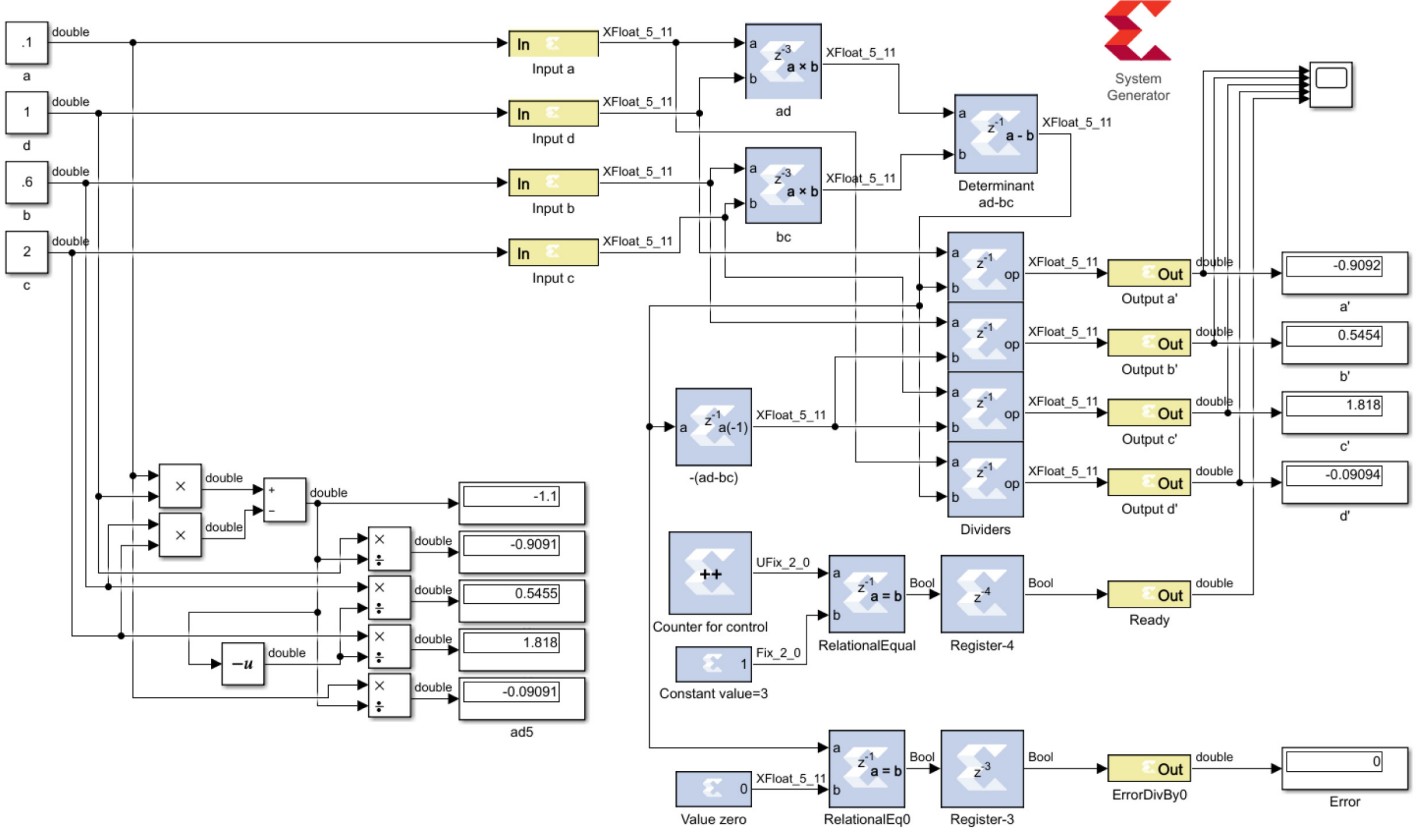

**Fig 3. Proposed architecture.** Architecture for computation of a 2 × 2 inverse matrix using the full unrolling technique.

enable a simple and fast determination of the inverse of the matrix. This prevents the evaluation of required resources depending on the algorithm, which, if scalability to different matrix sizes is considered, must be suitably modified. Nevertheless, the issues indicated here also arise in real-life situations, where the matrix size obtained is much larger, which might make it difficult to perceive these problems, such as in the examples shown in the section devoted to the operational equations of the first kind and their discretization.

The process for the implementation of the matrix inversion method is divided into three steps: 1) calculating the determinant, 2) calculating the co-factors and their division to generate the inverse matrix, and 3) defining control logic to indicate the end of the general calculation. The architecture is based on the fact that different data bus sizes are explored, and that is reflected in the use of more bits and interconnection cables. This is precisely the point of design exploration since it is necessary to know the impact of increasing precision and, consequently, increasing the number of interconnection cables.

The architecture has four input buses (the 4 elements of matrix $A$) and six output buses (the 4 elements of $A^{-1}$, and two pines, *Ready* and *ErrorDivBy*0, explained next). As we mentioned, the first step in the implementation of the matrix inversion method is the calculation of the determinant, which is done using the three upper shaded blocks, i.e., two multipliers and a subtractor. The second step is the calculation of co-factors (which requires the block for changing the sign of the determinant) and its division by the determinant (9 divisors). Finally, the third step is reflected by the modules in the lower part, where the counter is initiated by an external signal, the counter value is compared with a constant, and the comparator's signal is

**Table 2. Designs and used devices.**

| Design (name)—Format (number size) | IOBs | Spartan-7 | Artix-7 | Kintex-7 | Virtex-7 |
|---|---|---|---|---|---|
| Half precision (HP)—16 bits | 130 | x | x | x | x |
| Simple precision (SP)—32 bits | 258 | x | x | x | x |
| Double precision (DP)—64 bits | 514 | | | | x |
| Extra precision (EP)—79 bits | 634 | | | | x |

The number of pines is computed using IOBs = $8p + 2$, where $p$ is the precision size.

delayed by four clock cycles to generate flags indicating the process has completed (output *Ready*). The output error signal if the determinant is zero is *ErrorDivBy*0, indicating *A* does not have an inverse.

In this work, four designs using the same architecture were evaluated: HP, SP, DP, and EP. Each design is differentiated by its data buses, as previously mentioned (see Table 2). The standard for extra precision is defined using 80 bits. However, in this work, the design has a size of 79 bits due to restrictions of the *Vivado Design Suite 2019.2* since System Generator, which is a part of Vivado's suite, does not allow using more than 79 bits for representing floating-point numbers. These designs were not implemented on all devices, because the number of IOBs is insufficient in most of the used FPGA devices: *Artix-7* xc7a200tffg1156-1 and *Kintex-7* xc7k325tfbg900-1 have 500 IOBs, and *Spartan-7* xc7s100fgga676-1 has 400 IOBs. Only *Virtex-7* xc7vx1140tflg1930-1 has sufficient IOBs (1100).

For the proposed hardware architecture, the methodology focuses on four points: a) the effects of precision with errors in ill-conditioned matrices; b) the effects of architecture design and different types of precision; c) the effects of the hardware architecture implementation according to physical requirements of the embedded systems; d) the effects of different FPGA technologies.

Specifically, the following goals were covered with this work. i) To illustrate the effects of changes in precision, and to show that not all real numbers can be discretized, yet only a few that are limited by the number of bits; ii) to show the effects of truncation and ill-conditioning in matrices; iii) to define metrics and trade-off analysis; iv) to show the effects of ill-conditioned matrices, demonstrating the effect of noise; v) to define and explore trade-offs between precision and ill-conditioning; vi) to define and explore trade-offs between precision and use of hardware resources; vii) to define and explore trade-offs between precision and performance (throughput and efficiency) of the different FPGA technologies.

The following section shows the effects of different trade-offs: on the one hand, between truncation and precision; on the other hand, between hardware implementations (evaluated in terms of used hardware resources, throughput, and efficiency), and precision, where the effect of precision on ill-conditioned matrices was analyzed using *MATLAB R2019a*, the hardware architecture was developed using *System Generator 2019.2*, and *Vivado 2019.2* was used to implement the hardware architecture on different FPGA devices.

## Results

Trade-off analyses were carried out to obtain a $2 \times 2$ inverse matrix, where *MATLAB* was used to truncate to lower or greater precision, and the data bus size was adjusted to accommodate each precision level.

## Trade-off analysis: Truncation and precision

In this section, results from two case studies are used to show the effect of having lower or greater precision, where the mantissa and exponent in the floating-point representation are changed. In these analyses, the following matrix is considered: $A = \begin{pmatrix} 1 & 1 \\ 0 & \varepsilon \end{pmatrix}$ where $\varepsilon$ takes different positive small values less than $10^{-6}$, where $det(A) = \varepsilon$, and the inverse of matrix $A$ is $A^{-1} = \begin{pmatrix} 1 & -1/\varepsilon \\ 0 & 1/\varepsilon \end{pmatrix}$. We consider the linear system $AX = B$, for the input data $B = \begin{pmatrix} 1 \\ 1 \end{pmatrix}$, whose exact solution is $\begin{pmatrix} 1 - 1/\varepsilon \\ 1/\varepsilon \end{pmatrix}$, which is denoted by $X_{exact}$. In this case, if $\varepsilon$ is very small, $A$ is a nearly singular matrix because $det(A)$ is approximately zero. This produces instability in solving the system of equations $AX = B$ when $\varepsilon$ is truncated (or rounded) to $s$ digits because the condition number of matrix $A$ is a very large number, as shown in Table 3. For example, if we consider the system of equations $AX = B$, with $\varepsilon = (1/3) \times 10^{-7}$, and if $\varepsilon$ is truncated to 4 digits, there is no solution to the corresponding approximate system using the method proposed here (although there is an infinite number of solutions using the algebraic method). If $\varepsilon$ is truncated to 8 digits, the approximate solution to the system may be far from the exact solution to the original system, as shown in Table 3. The condition number is used to analyze the numerical instability that appears when solving the system of equations, which is defined as follows:

**Definition 1**. *Let* $\||\cdot\||$ *be a subordinate norm and* $A \in \mathcal{M}_{m*n}$ *be an invertible matrix. The number* $coad(A) = \||A\|| \; \||A^{-1}\||$ *is called the condition number of the matrix associated to the norm* $\||A\||$. *If we consider the Euclidean norm on* $\Re^d$, *the subordinate norm* $\||A\||_2$ *is defined in the following form:* $\||A\||_2 = \sqrt{\rho(A^t A)}$ *where* $\rho(A) := max\{|\lambda|: \lambda \text{ is an eigenvalue of } A\}$ *denotes the spectral radius of* $A$, *and a subordinate norm* $\||\cdot\||$ *for a matrix* $A$ *is defined as* $\||A\|| = sup_{x \in \Re^d} \frac{\|Ax\|}{\|x\|}$.

**Table 3. Numeric results by truncating $\varepsilon$ to too few digits.**

| $\varepsilon$ | $(1/3) \times 10^{-7}$ | $(1/3) \times 10^{-15}$ | $(1/3) \times 10^{-31}$ |
|---|---|---|---|
| **Truncation digits ($s$)** | 8 | 16 | 32 |
| $\varepsilon_{approx}$ | $3 \times 10^{-8}$ | $3 \times 10^{-16}$ | $3 \times 10^{-32}$ |
| cond($A$) | $6.0000 \times 10^{7}$ | $6.0000 \times 10^{15}$ | $6.0000 \times 10^{31}$ |
| cond($A_{approx}$) | $6.6667 \times 10^{7}$ | $6.6667 \times 10^{15}$ | $6.6667 \times 10^{31}$ |
| $A^{-1}$ | $\begin{pmatrix} 1 & -3.0000 \times 10^{7} \\ 0 & 3.0000 \times 10^{7} \end{pmatrix}$ | $\begin{pmatrix} 1 & -3.0000 \times 10^{15} \\ 0 & 3.0000 \times 10^{15} \end{pmatrix}$ | $\begin{pmatrix} 1 & -3.0000 \times 10^{31} \\ 0 & 3.0000 \times 10^{31} \end{pmatrix}$ |
| $A^{-1}_{approx}$ | $\begin{pmatrix} 1 & -3.3333 \times 10^{7} \\ 0 & 3.3333 \times 10^{7} \end{pmatrix}$ | $\begin{pmatrix} 1 & -3.3333 \times 10^{15} \\ 0 & 3.3333 \times 10^{15} \end{pmatrix}$ | $\begin{pmatrix} 1 & -3.3333 \times 10^{31} \\ 0 & 3.3333 \times 10^{31} \end{pmatrix}$ |
| $X_{exact}$ | $(-3.0000 \times 10^{7}, 3.0000 \times 10^{7})^t$ | $(-3.0000 \times 10^{15}, 3.0000 \times 10^{15})^t$ | $(-3.0000 \times 10^{31}, 3.0000 \times 10^{31})^t$ |
| $X_{approx}$ | $(-3.3333 \times 10^{7}, 3.3333 \times 10^{7})^t$ | $(-3.3333 \times 10^{15}, 3.3333 \times 10^{15})^t$ | $(-3.3333 \times 10^{31}, 3.3333 \times 10^{31})^t$ |
| AE($X_{approx}$, $X_{exact}$) | $4.7140 \times 10^{6}$ | $4.7140 \times 10^{14}$ | $4.7140 \times 10^{30}$ |
| RE($X_{approx}$, $X_{exact}$) | 0.1111 | 0.1111 | 0.1111 |

$\varepsilon$ is truncated to 8, 16, and 32 digits.

Estimating the condition number is a topic of interest for the scientific community since it is important when describing the effectiveness of new preconditioners or selecting adequate preconditioners [19].

For approximate values of $\varepsilon$, we consider solving the previous linear system by approximating $A$ as follows: $A_{approx} = \begin{pmatrix} 1 & 1 \\ 0 & \varepsilon_{approx} \end{pmatrix}$, where $\varepsilon_{approx}$ is a truncation of $\varepsilon$ to $s$ digits. In this case, $det(A_{approx}) = \varepsilon_{approx}$.

As a didactic example of the effect of truncation on an inverse matrix calculation, we have:

**Example.** We consider the matrix $A_{\epsilon_1} = \begin{pmatrix} 1 & 1 \\ 0 & \epsilon_1 \end{pmatrix}$, where $\epsilon_1 = 0.011$; and

$A_{\epsilon_2} = \begin{pmatrix} 2 & 1 \\ 0 & \epsilon_1 \end{pmatrix}$, where $\epsilon_2 = 0.01$. The inverse matrices are

$A_{\epsilon_1}^{-1} = \frac{1}{\epsilon_1}\begin{pmatrix} \epsilon_1 & -1 \\ 0 & 1 \end{pmatrix} \approx \begin{pmatrix} 1 & -90.90 \\ 0 & 90.90 \end{pmatrix}$, and $A_{\epsilon_1}^{-1} = \frac{1}{\epsilon_1}\begin{pmatrix} \epsilon_2 & -2 \\ 0 & 1 \end{pmatrix} \approx \begin{pmatrix} 2 & -100 \\ 0 & 100 \end{pmatrix}$, respectively. We can see that the inverse matrices are significantly different, even if $\epsilon_1$ and $\epsilon_2$ are similar. The effect of truncation on the solution to these systems is illustrated as follows: we consider the system $AX = B$ with matrices given by $A_{\epsilon_1}$ and $A_{\epsilon_2}$, where the right side is

$B = \begin{pmatrix} 1 \\ 1 \end{pmatrix}$. The respective solutions are $\begin{pmatrix} x_1^1 \\ x_2^1 \end{pmatrix} = \begin{pmatrix} -89.90 \\ 90.90 \end{pmatrix}$ and $\begin{pmatrix} x_1^2 \\ x_2^2 \end{pmatrix} = \begin{pmatrix} -99.90 \\ 100 \end{pmatrix}$, and the Euclidean distance between both solutions is 13.52.

In the following, we present two different case studies with respect to truncation of $\varepsilon$. In Case Study 1, $\varepsilon$ is truncated to too few decimal places; in Case Study 2, $\varepsilon$ is truncated to an adequate number of decimal places (which is defined in terms of the desired precision, depending on the value of the smallest input number, in this case, $\varepsilon$).

**Case Study 1 (too few decimal digits):** Table 3 shows the errors in the solution to the system $AX = B$ when $\varepsilon$ is truncated to $s$ digits. In this table, the Euclidean distance between $X$ and $X_{approx}$ is $4.7140 \times 10^6$ when $\varepsilon = \frac{1}{3} \times 10^{-7}$. However, the absolute error (AE) between the exact and approximate solutions is $4.7140 \times 10^6$, which illustrates the level of numerical instability. The case of $\varepsilon = \frac{1}{3} \times 10^{-31}$ is even more dramatic since the Euclidean distance is $4.7140 \times 10^{30}$, which presents a great problem for applications, that is, the numerical instability could have important effects on the solution. Notice that the condition number grows, which shows that errors can be amplified. Similar results are obtained when $\varepsilon$ takes smaller values. For example, when $\varepsilon = \frac{1}{3} \times 10^{-63}$ is truncated to 64 digits, an AE of $4.7140 \times 10^{62}$ is obtained and the error in the elements of the approximate inverse of $A$ increases, which could be a very large error for some applied problems.

As we can observe in this case study and in general, ill-conditioning in a matrix can lead to a situation wherein the solution to a system with erroneous data can be too far from the solution with exact data (as shown in Table 3). This fact is independent of the numeric method used to compute the solution to the system, since similar results are obtained for various methods.

**Remark:** The ill-conditioning of a matrix does not depend on errors (e.g., by rounding up and down, which includes truncation) since it is inherent in the matrix itself. Ill-conditioning amplifies any errors that appear in the system, such as rounding errors or errors inherent in the measurement devices. Conversely, truncation is occasionally used for saving hardware or

**Table 4. Numeric results when precision of $\varepsilon = (1/3) \times 10^{-7}$ was increased to find an adequate number of decimal digits.**

| Truncation digits ($s$) | 8 (inadequate) | 16 (adequate) | 32 (adequate) |
|---|---|---|---|
| $\varepsilon_{approx}$ | $3 \times 10^{-8}$ | $333333333 \times 10^{-16}$ | $3333\ldots3333 \times 10^{-32}$ |
| cond($A$) | $6.0000 \times 10^7$ | $6.0000 \times 10^7$ | $6.0000 \times 10^7$ |
| cond($A_{approx}$) | $6.6667 \times 10^7$ | $6.0000 \times 10^7$ | $6.0000 \times 10^7$ |
| $A_{approx}^{-1}$ | $\begin{pmatrix} 1 & -3.3333 \times 10^7 \\ 0 & 3.3333 \times 10^7 \end{pmatrix}$ | $\begin{pmatrix} 1 & -3.0000 \times 10^7 \\ 0 & 3.0000 \times 10^7 \end{pmatrix}$ | $\begin{pmatrix} 1 & -3.0000 \times 10^7 \\ 0 & 3.0000 \times 10^7 \end{pmatrix}$ |
| $X_{approx}$ | $(-3.3333 \times 10^7, 3.3333 \times 10^7)^t$ | $(-3.0000 \times 10^7, 3.0000 \times 10^7)^t$ | $(-3.0000 \times 10^7, 3.0000 \times 10^7)^t$ |
| AE($X_{approx}, X_{exact}$) | $4.7140 \times 10^6$ | 0.0424 | 0 |
| RE($X_{approx}, X_{exact}$) | 0.1111 | $1.0000 \times 10^{-9}$ | 0 |

The number of digits is considered adequate if AE and RE are small enough (although the term 'small' depends on the specific problem).

software resources, or both. Hence, we analyzed the effects of truncation with ill-conditioned matrices.

**Case Study 2 (adequate number of decimal digits)**: Table 4 considers the solution to the system $AX = B$, when $\varepsilon = (1/3) \times 10^{-7}$, whose exact solution is $\begin{pmatrix} -3.0000 \times 10^7 \\ 3.0000 \times 10^7 \end{pmatrix}$; in this case, $A^{-1} = \begin{pmatrix} 1 & -3.0000 \times 10^7 \\ 0 & 3.0000 \times 10^7 \end{pmatrix}$. It is observed that, by increasing the precision of $\varepsilon$ to $s$ digits, the approximate solution $X_{approx}$ reaches closer to the exact solution $X_{exact}$. For example, when $\varepsilon = (1/3) \times 10^{-7}$ is approximated with 8 digits, the AE and relative error (RE) in the approximate solution $x_{approx}$ are $4.7140 \times 10^6$ and 1.1111, respectively, which are greater than the AE and RE when $\varepsilon$ is approximated with 16 digits, where the AE and RE are 0.0424 and $1.0000 \times 10^{-9}$, respectively. In this case, we get a good approximation of the inverse matrix $A^{-1}$. Moreover, if $\varepsilon = (1/3) \times 10^{-7}$ is truncated to 32 digits, both AE and RE are zero (also shown in Table 4). Observe that precision in the number of digits will depend on the specific problem to be solved, e.g., in [20], some parameters of the mathematical model are in the interval $[10^{-20}, 10^{20}]$.

From this case study, one finds it is feasible to determine the adequate number of digits $s$ that can reduce calculation errors. In this case, given that $\varepsilon = (1/3) \times 10^{-7}$, the adequate number is 16 (in terms of precision). Nevertheless, as we will see in the following analysis, there is an inherent cost in increasing precision that must be considered when determining if such a solution is viable (e.g., more than 16 digits might consume unnecessary hardware resources).

## Trade-off analysis: Hardware implementations and performance

Synthesis results for the proposed hardware architecture are presented in this section. For the purpose of validation and prototyping, this architecture is synthesized, mapped, placed and routed using *Spartan-7*, *Artix-7*, *Kintex-7*, and *Virtex-7* FPGAs; *Vivado 2019.2* was used as the design tool. The implemented architecture was simulated and verified considering real-time operation conditions by using design conformance test data.

Several metrics are considered for evaluating these architectures and analyzing different trade-offs. Some metrics are provided by the tool, such as the number of slices, look-up tables (LUTs), flip-flops (FFs), block random access memories (BRAMs), digital signal processors (DSPs), and IOBs, along with the minimum clock period and power consumption. In addition,

other metrics can be obtained by using certain equations; these metrics are throughput and efficiency.

Throughput depends on the number of bits processed per second (bps), where the clock frequency is obtained by implementing the architecture on different FPGA technologies (see Eq (10)). The second metric is implementation efficiency, which is a measure of these types of hardware implementations and is defined as the ratio between the reached throughput and the physical area of the device (i.e., amount of hardware resources) used for implementation, such as the number of LUTs that each implementation consumes (bps/LUT) and the number of flip-flops (FFs) used (bps/FF) (see Eq (11)).

$$Throughput = Data\_input \times Clock\_frequency/Clock\_cycles, \qquad (10)$$

$$Efficiency = Throughput/Area. \qquad (11)$$

Considering these metrics, the hardware architecture was implemented using four technologies (*Artix-7*, *Kintex-7*, *Spartan-7*, and *Virtex-7*) for the MP and SP versions (see Tables 5 and 6, respectively), and one technology (*Virtex-7*) for the DP and EP versions (see Tables 7 and 8, respectively).

Figs 4 and 5 show the results when the hardware architecture was implemented on different FPGAs. These metrics were obtained from the *Vivado 2019.2* tool for each FPGA used. Different available numbers of LUTs, look-up table random access memory (LUTRAM), FFs, DSPs and IOBs are presented. The critical path time is also shown, which defines the minimum clock period and, consequently, the maximum clock frequency. Next, two different analyses are considered: 1) horizontal, where the same designs (HP, SP, DP, or EP) are compared using different devices, and 2) vertical, where different designs are compared, which implies different required precision levels.

The first analysis shows that the required number of LUTs for each design is highly similar, regardless of the FPGA being used. The amount of LUTRAM (plot not shown), and the number of FFs, DSPs, and IOBs are the same for all FPGAs. The clock periods in the *Artix-7* and *Spartan-7* FPGAs are similar, while the first clock period is larger for *Kintex-7* and *Virtex-7*. For example, the HP design requires 28.2 ns (*Spartan-7*) versus 17.33 ns (*Kintex-7*). This

**Table 5. Implementation results for the MP-based hardware architecture.**

| Metric (Hw) | MP | | | |
|---|---|---|---|---|
| | Artix-7 xc7a200tffg1156-1 | Kintex-7 xc7k325tfbg900-1 | Spartan-7 xc7s100fgga676-1 | Virtex-7 xc7vx1140tflg1930-1 |
| LUT | 1045 | 1031 | 1044 | 1030 |
| LUTRAM | 2 | | | |
| FF | 214 | | | |
| DSP | 4 | | | |
| IOB | 131 | | | |
| Data size (bits) | 64 | | | |
| Latency (clk cycles) | 6 | | | |
| Time (ns) | 27.346 | 17.33 | 28.2 | 17.275 |
| Throughput (Mbps) | 390.06 | 615.50 | 378.25 | 617.46 |
| Efficiency (Mbps/LUT) | 0.373266164 | 0.596996102 | 0.362308995 | 0.599478267 |
| Efficiency (Mbps/FF) | 1.82272496 | 2.876182156 | 1.767526126 | 2.88533932 |
| Power (W) | 0.088 | 0.087 | 0.087 | 0.087 |

**Table 6. Implementation results for the SP-based hardware architecture.**

| | SP | | | |
|---|---|---|---|---|
| Metric (Hw) | Artix-7 xc7a200tffg1156-1 | Kintex-7 xc7k325tfbg900-1 | Spartan-7 xc7s100fgga676-1 | Virtex-7 xc7vx1140tflg1930-1 |
| LUT | 3478 | 3456 | 3476 | 3456 |
| LUTRAM | 2 | | | |
| FF | 308 | | | |
| DSP | 6 | | | |
| IOB | 259 | | | |
| Data size (bits) | 128 | | | |
| Latency (clk cycles) | 6 | | | |
| Time (ns) | 64.204 | 40.63 | 66.976 | 40.231 |
| Throughput (Mbps) | 166.14 | 262.53 | 159.26 | 265.14 |
| Efficiency (Mbps/LUT) | 0.047768 | 0.07596406 | 0.045817327 | 0.076717451 |
| Efficiency (Mbps/FF) | 0.539406184 | 0.852375945 | 0.517081262 | 0.860829575 |
| Power (W) | 0.198 | 0.202 | 0.193 | 0.203 |

**Table 7. Implementation results for the DP-based hardware architecture.**

| | DP |
|---|---|
| Metric (Hw) | Virtex-7 xc7vx1140tflg1930-1 |
| LUT | 13381 |
| LUTRAM | 2 |
| FF | 628 |
| DSP | 22 |
| IOB | 515 |
| Data size (bits) | 256 |
| Latency (clk cycles) | 6 |
| Time (ns) | 126.047 |
| Throughput (Mbps) | 84.62 |
| Efficiency (Mbps/LUT) | 0.006 |
| Efficiency (Mbps/FF) | 0.135 |
| Power (W) | 0.611 |

**Table 8. Implementation results for the EP-based hardware architecture.**

| | EP |
|---|---|
| Metric (Hw) | Virtex-7 xc7vx1140tflg1930-1 |
| LUT | 19168 |
| LUTRAM | 2 |
| FF | 746 |
| DSP | 34 |
| IOB | 635 |
| Data size (bits) | 512 |
| Latency (clk cycles) | 6 |
| Time (ns) | 176.846 |
| Throughput (Mbps) | 60.32 |
| Efficiency (Mbps/LUT) | 0.003 |
| Efficiency (Mbps/FF) | 0.081 |
| Power (W) | 0.827 |

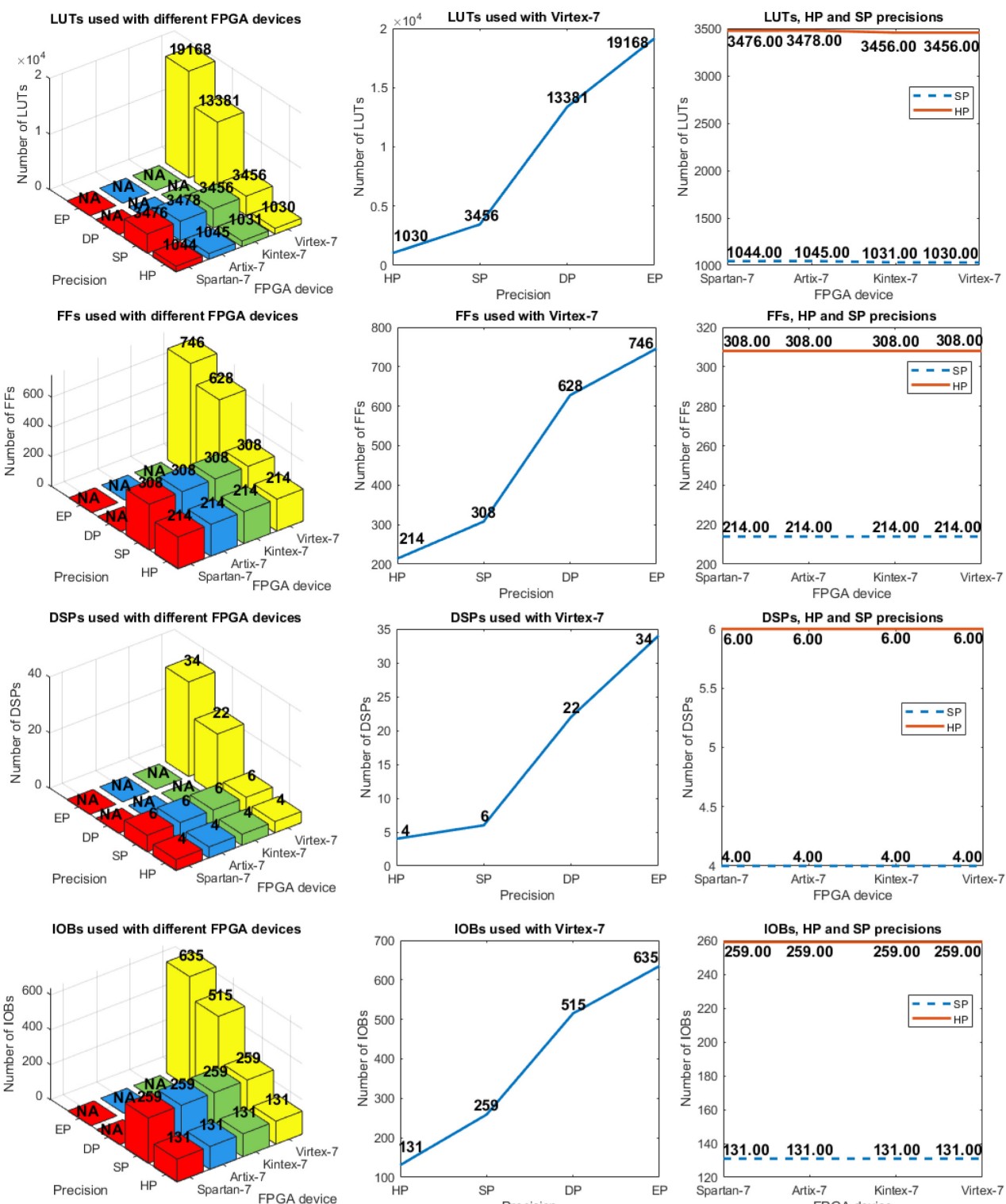

**Fig 4.** (left) LUTs, FFs, DSPs, and IOBs used with different FPGA devices and precision levels (these are presented in inverse order for better visualization); (center) Similar analysis with *Virtex-7* FPGA with different precision levels; (right) Similar analysis for the comparison of precision with HP and SP using different FPGA devices.

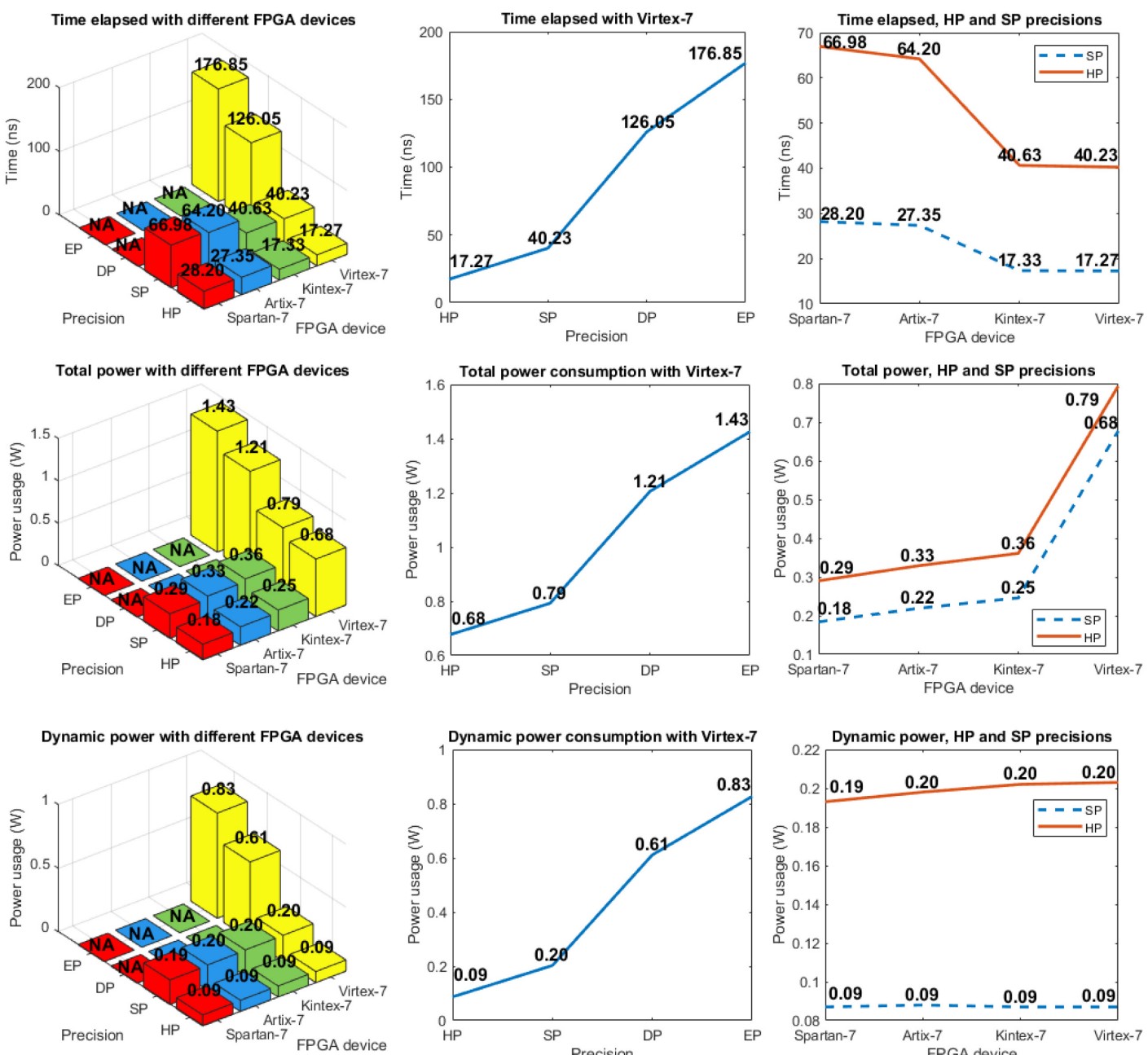

**Fig 5.** (left) Time, total power, and dynamic power used with different FPGA devices and precision levels (these are presented in inverse order for better visualization); (center) Similar analysis with *Virtex-7* FPGA with different precision levels; (right) Similar analysis for the comparison of precision with HP and SP using different FPGA devices.

means that the same design requires longer computation time for the same operations. In the case of device power consumption, larger devices (*Virtex-7*) consume more power, although each design requires a similar amount of hardware resources, which requires highly similar power (this is called dynamic power consumption). For example, the SP design consumes from 193 mW (*Spartan-7*) to 203 mW (*Virtex-7*).

**Table 9. Growth factor of the SP, DP, and EP designs.**

| Concept | SP | DP | EP | SP | DP | EP |
|---|---|---|---|---|---|---|
| | Growth factor with respect to the immediate lower precision level | | | Growth factor with respect to the HP design | | |
| LUTs | 3.3 | 3.8 | 1.4 | 3.3 | 12.9 | 18.6 |
| FFs | 1.4 | 2.0 | 1.1 | 1.4 | 2.9 | 3.4 |
| DSPs | 1.5 | 3.6 | 1.5 | 1.5 | 5.5 | 8.5 |
| Dynamic power | 2.2 | 3.0 | 1.3 | 2.2 | 6.7 | 9.2 |
| Time | 2.3 | 3.1 | 1.4 | 2.3 | 7.2 | 10.2 |

Implementations on the *Virtex-7* FPGA.

The HP design considers 16-bit numbers, whereas SP, DP, and EP consider 32, 64, and 79-bit numbers (remember that, in this work, EP design is reduced to 79-bit numbers due to the restrictions in *System Generator 2019.2*).

With respect to the second analysis, we highlight that the required number of LUTs in the SP design is approximately a factor 3.3 larger than that in the HP design, whereas data buses pass from 16 bits in HP to 32 bits in SP. This change in the number of LUTs is a factor 3.87 larger when the DP and SP designs are compared (passing from 32 bits in SP to 64 bits in DP); the change for the EP design is a factor 1.43 larger compared to the DP design (passing from 64 bits in DP to 79 bits in EP). In a minor proportion, this occurs because the number of required FFs increased, whereas the amount of LUTRAM is constant (2) for all designs (this is why its plot is not shown). Regardless of the FPGA device, the number of DSPs increases from 4 for the HP design to 6 for the SP design; however, it increases to 22 for the DP design, and to 34 for the EP design. Likewise, dynamic power consumption increases from 87 mW for the HP design to 203 mW (SP), 611 mW (DP), and 827 mW (EP). Finally, considering implementation on *Virtex-7* and the corresponding clock period, the SP, DP, and EP designs require factor 2.33, 7.29, and 10.21 more time than the HP design. This last metric has a direct effect on performance. The different growth factors in requirements (LUTs, FFs, DSPs, dynamic power, and time), with the different precision levels (HP, SP DP, and EP) implemented on Virtex-7, are presented in Table 9.

The trade-off between physical resources and precision shows that increasing the precision by a factor 2 requires, in certain cases, more than a factor 2 or 3 more hardware resources and power consumption, which means that their growth seems to be not linear.

Fig 6 shows how the *Vivado* tool fits the SP design into different devices. These representations show the proportion of hardware required to implement the design and the remaining hardware on the device. The *Virtex-7* device is the largest, thus it consumes the most total power.

Regarding the trade-off between performance and precision, it is necessary to know the data block size and latency. The HP, SP, DP, and EP designs process four 16, 32, 64, and 79 bit numbers, respectively. Hence, each design processes, respectively, 64, 128, 256, and 316 bits when computing a $2 \times 2$ inverse matrix. Moreover, all designs have a latency of 6 clock cycles because 4 clock cycles are used to compute the determinant (3 clock cycles for two parallel multipliers and 1 clock cycle for the subtractor) and 2 clock cycles are used to compute the cofactors and elements of the inverse matrix (1 clock cycle for changing the sign of the determinant and 1 clock cycle for divisors) (remember Fig 3).

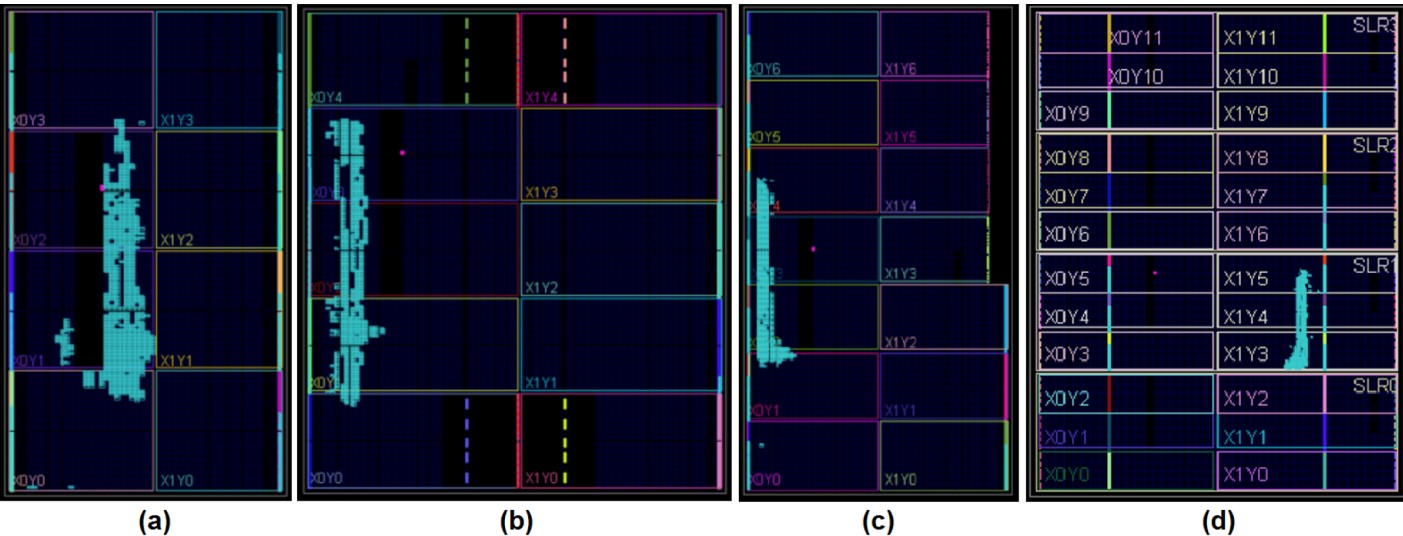

**Fig 6. Fitting the SP design into the different devices.** (a) *Spartan-7*, (b) *Artix-7*, (c) *Kintex-7*, and (d) *Virtex-7*.

A performance evaluation of the hardware architecture on the FPGA devices (shown in Fig 7) considers throughput (Eq (10)) and two efficiencies: a) throughput/(number of LUT) and b) throughput/(number of FF), both using Eq (11).

Throughput shows the amount of data (bits) a given implementation processes per unit of time (in our case, per second). The throughput in the HP design is higher on the *Kintex-7* and *Virtex-7* devices, improving 1.43 times when *Virtex-7* is compared against *Spartan-7*, which is achieved because the *Virtex-7* technology delivers better conditions. Throughput decreases when precision increases. For example, the throughput with *Virtex-7* was 617.46 Mbps for HP, 265.14 Mbps for SP, 84.62 Mbps for DP, and 60.32 Mbps for EP. This metric indicates which technology processes the largest amount of data.

Efficiency shows which implementation uses the hardware resources (in our case, LUTs and FFs) in a better way, while processing a certain amount of data. The HP design requires practically the same number of LUTs and FFs in the *Kintex-7* and *Virtex-7* FPGAs, in which the highest efficiency is seen. However, efficiency decreases when precision increases. For example, for the *Virtex-7* FPGA, efficiency decreases by 87.14% when precision increases from 16 to 32 bits, by another 92.20% when precision increases from 32 to 64 bits, and by another 50.0% when precision increases from 64 to 79 bits.

Trade-off between performance and precision shows that both throughput and efficiency drastically decrease when precision increases.

The proposed architecture has a parallel and specific structure, which is scalable because it is based on invertible matrices using the adjoint method. If new mathematical models require other matrix sizes (in general a size of $n \times n$), more modules are required according to the operation count, for example, $\left[ n! + n^2 + 3\left(\frac{n(n+1)}{2} - 3\right)\right]$ multipliers, and $\left[\frac{n!}{2} + \left(\frac{n(n+1)}{2} - 3\right)\right]$ adders, where $\frac{n!}{2}$ determinants of size $2 \times 2$ are calculated. However, there are other hardware-design techniques that are more suitable for reducing modules, which will be explored in future works.

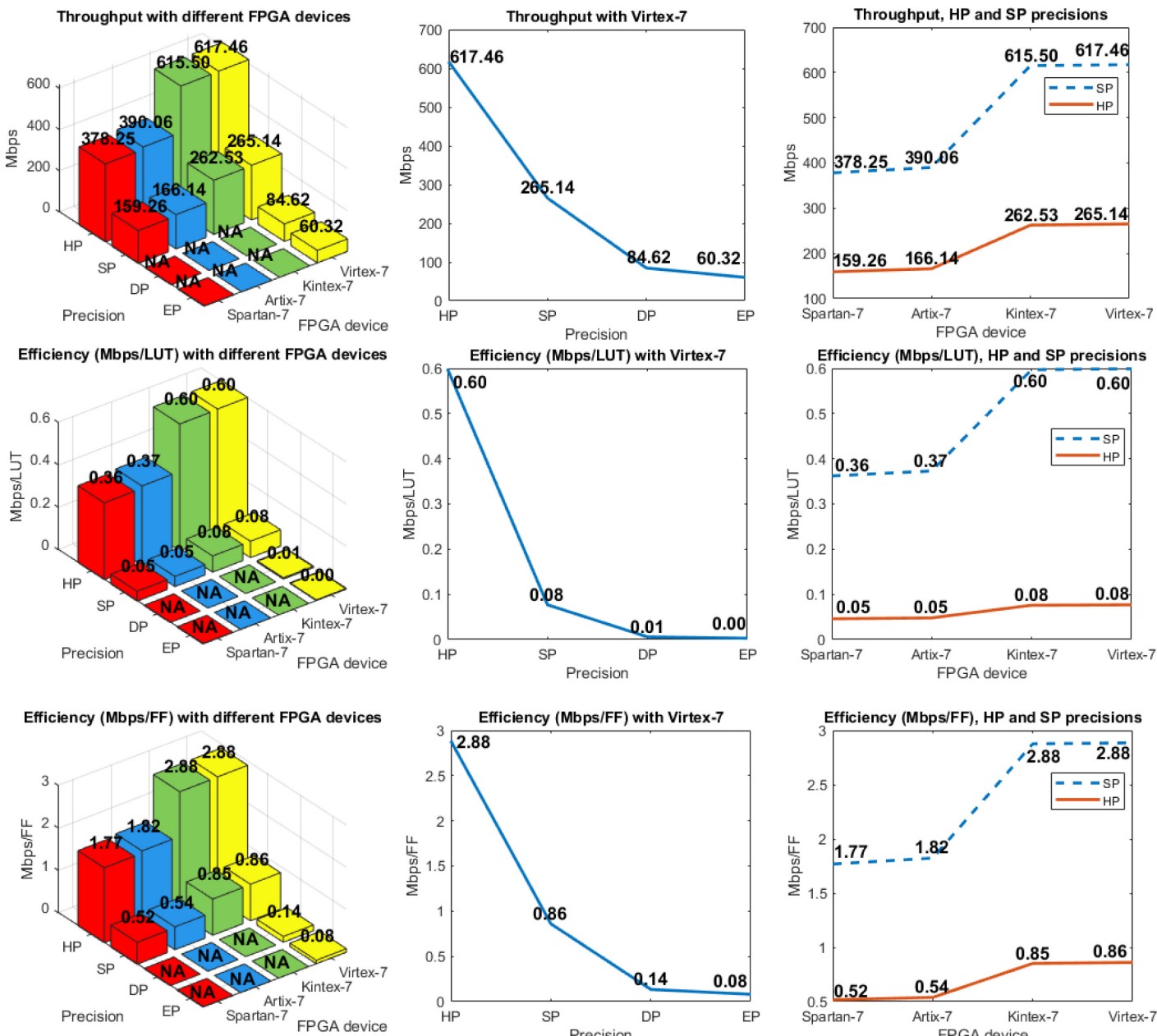

**Fig 7. Throughput and efficiency (Mbps/LUT and Mbps/FF, respectively).** (left) Different FPGA devices and precision levels; (center) *Virtex-7* with different precision levels; (right) comparison of precision in the HP and SP designs with using different FPGAs. Throughput and efficiency were computed using Eqs (10) and (11), respectively.

## Discussion

As was previously mentioned, matrices appear in several problems involving systems of algebraic equations with ill-conditioned matrices. Regularization methods are used to handle ill-conditioning, among which Tikhonov's is the most famous. The condition number of a matrix works as a measure of the system's sensitivity to errors. These matrices can appear when some

operational equations are discretized, yielding a system of higher-order equations. This implies that significant resources can be required for implementation.

Specifically, in this article, we present a hardware architecture with four designs and different precision levels (HP, SP, DP, and EP), where each is implemented on four different FPGA technologies, thus trade-off analysis considers twelve implementations. Each of the four designs has a fixed type of data, and other types must be cast to the requirements of some selected design. In this process, rounding and truncation are necessary. However, critical path, throughput, efficiency, and amount of hardware resources do not change because they are dependent, fixed, and inherent characteristics of the implementation on some FPGA. Hence, once the architecture is implemented, the amount of resources needed for calculations becomes constant and does not depend on the type of the input data (i.e., if the type of the input data is different from the type implemented in the architecture, the former is converted through a casting operation, by filling, truncating, or rounding the number, depending on the case). For example, if the architecture works with 32-bit floating-point numbers, and the inputs are 16-bit integers, they must be converted through a casting operation to the equivalent numbers (or the closest to them) before being entered into the FPGA.

Certain aspects of the results presented in this work are straightforward to researchers from areas such as electronics, mathematics, physics, or computer science, i.e., a mathematician is conscious that ill-conditioning will certainly provoke a divergence in the calculation results. In such cases it will be necessary to apply a regularization method (which is not included in the FPGA implementations presented in this work) or, at least, consider the precision of the data used in the calculation of the results while accounting for inherent errors in the data [3, 5, 7, 11, 19–21]. On the other hand, a computer or electronic engineer is conscious of the costs associated with increasing precision in various calculation (or other aspects, such as the matrix size), such as the changes (not necessarily proportional, as we have shown) in processing time, energy, hardware resources, and memory space [22–26].

Several works have addressed ill-conditioning using classical or novel techniques to solve real-life problems [3, 5, 7, 11, 19–21]; however, in several cases, the problems stated here (limited resources and ill-conditioning) must be considered to avoid the following complications and/or limitations:

- creating models despite having limited resources where, although ill-conditioning might be taken into account, the costs associated with the increased precision are not clear and/or are minimized, making the modeled solution difficult to apply to real-life problems;

- introducing restrictions on the solution, to avoid ill-conditioning at certain processing stages, also limiting with this the applicability of such a solution; and

- increasing precision, expecting that this will improve calculation results (by reducing error), ignoring or disregarding ill-conditioning, making the solution more sensitive to errors.

## Comparisons

On the topic of FPGAs used for matrix inversion, Karkooti et al [23] presented an FPGA implementation of the QR decomposition-based recursive least square (RLS) algorithm using a *Xilinx Virtex-4* FPGA, yielding a 4 × 4 matrix of complex numbers. The authors claim their architecture is easily scalable to other matrix sizes, with the advantages of being less affected by impulsive noise and making efficient use of bandwidth. They reported a throughput of 0.13M updates per second on 14 bits for mantissa, 6 bits for the exponent of floating-point numbers, and 1 sign bit, requiring 9,117 slices, 22 DSP48, 9 BRAMs and 309 IOBs.

The work on matrix inversion by Irturk et al. [24] is also based on QR decomposition. The authors used the *Virtex-4* FPGA for $4 \times 4$ matrix inversion and signal equalization, where 19, 26, and 32-bit precision levels were used (fixed-point arithmetic). They reported, for $4 \times 4$ matrices: 1) 19-bit implementation requires 3,528 LUTs, 1,731 FFs, 1 BRAM, 12 DSP48, and 2,415 slices, presenting a throughput of 0.18 M updates per second; 2) 26-bit implementation requires 7,486 LUTs, 3,276 FFs, 1 BRAM, 12 DSP48, and 4,656 slices, presenting a throughput of 0.14 M updates per second; 3) 32-bit implementation requires 8,804 LUTs, 4,208 FFs, 1 BRAM, 12 DSP48, and 5,640 slices, presenting a throughput of 0.11 M updates per second. For 19-bit precision, the authors illustrated the scalability of their method, presenting results for 1) $6 \times 6$ matrices: 7,820 slices, 2 BRAMs, and 18 DSP48, reporting a throughput of 0.07M; 2) $8 \times 8$ matrices: 11,761 slices, 4 BRAMs, and 24 DSP48, with a throughput of 0.03M.

Arias-García et al. [26] present mean-error analysis for computing $n \times n$ matrix inversion, where $n$ ranged from 5 to 120. They used a *Virtex-5* device for matrix inversion with different precision levels (32, 40 and 64 bits), although they do not present implementation results for the full architecture, only for the main modules.

Additionally, Kumar et al [22] implemented the adjoint matrix method using very high speed integrated circuit hardware description language (VHDL) and an *Altera DE1* FPGA, computing a $3 \times 3$ matrix of real numbers. According to the authors, the major limitations of their implementation are i) the implemented method is not suitable for higher-order matrices; ii) inputs and outputs are in a different format; therefore, the property $(A^{-1})^{-1} = A$ cannot be verified. Although authors do not present results on throughput or efficiency, they report 5,501/18,752 (29%) logic elements (LEs), 80/18,752 (<1%) logic registers (LRs), 36/315 (11%) I/O pins, and 24/52 (46%) embedded multipliers.

Finally, Ruan [25] presents a C-based implementation for generating pipelined architectures in VHDL for a given $n \times n$ matrix. He reports implementations for 1) a $32 \times 32$ matrix: 8,669 LUTs, 9,024 FFs, 60 DSPs, 44 BRAM18K, operating at 351 MHz, and a latency of 203,676 clock cycles; 2) $20 \times 20$ matrix: 10,159 LUTs, 10,255 FFs, 60 DSPs, 22 BRAM18K, operating at 330 MHz, and a latency of 66,063 clock cycles; 3) $16 \times 16$ matrix: 9,033 LUTs, 10,097 FFs, 60 DSPs, 22 BRAM18K, operating at 313 MHz, and a latency of 45,861 clock cycles; 4) $8 \times 8$ matrix: 8,889 LUTs, 9,974 FFs, 60 DSPs, 22 BRAM18K, operating at 330 MHz, and a latency of 10,765 clock cycles.

One can see that, except for the results from [24, 26], most authors compare their results with those from other authors without performing an in-depth analysis of trade-offs. One of these prior studies even examines implementations with non-standard precision levels [23]. One can also see that some prior studies focused on scalability (with respect to matrix dimension), which is a basic attribute of a system applicable to real-life problems. Nevertheless, a thorough analysis of the variations and effects in hardware with certain precision requirements, like that presented in this paper, or in the presence of anomalies, such as ill-conditioning, has not been performed previously, to the best of our knowledge. Without this kind of analysis, neither increasing precision, nor scaling an architecture, might be satisfactory solutions. Conversely, this will likely generate more problems in terms of obtaining inadequate results and unnecessary consumption of additional resources. A comparison of results from earlier studies is shown in Table 10.

## Conclusions

In this study, the problem of ill-conditioning and the trade-offs involved in increasing computational precision in hardware or software was investigated from two perspectives. On the one hand, the analysis of ill-conditioned matrices, using the didactic examples and case studies

**Table 10. Comparative table with respect to other FPGA implementations of matrix inversion methods.**

| Year | Reference | Algorithm | Device | Matrix size | Details (slices, LUTs, FFs, Mbps, IOBs, etc) |
|---|---|---|---|---|---|
| 2005 | Karkooti et al. [23] | QRD-RLS | Virtex-4 | $4 \times 4$ | 9117 Slices, 22 DSP48, 9 BRAMs, 309 IOBs, 0.13M updates |
| 2009 | Irturk et al. [24] | QRD | Virtex-4 | $4 \times 4$ | 19-bit numbers, 3,528 LUTs, 1,731 FFs, 1545BRAM, 12 DSP48, 2,415 slices, 0.18 M updates |
| | | | | $4 \times 4$ | 26-bit numbers, 7,486 LUTs, 3,276 FFs, 1 BRAM, 12 DSP48, 4,656 slices, 0.14 M updates |
| | | | | $4 \times 4$ | 32-bit numbers, 8,804 LUTs, 4,208 FFs, 1 BRAM, 12 DSP48, 5,640 slices, 0.11 M updates |
| 2012 | Arias-García et al. [26] | Gauss-Jordan | Virtex-5 | $5 \times 5$–$120 \times 120$ | Modules are individually implemented |
| 2014 | Kumar et al. [22] | Adjoint Matrix, Cayley-Hamilton | Altera DE1 | $3 \times 3$ | 5501 LEs, 80 LRs, 36 IOBs, 24 multipliers |
| 2017 | Ruan [25] | QRD-MGS | Virtex-4 | $32 \times 32$ | 8,669 LUTs, 9,024 FFs, 60 DSPs, 44 BRAM18K, 351 MHz, 203,676 clock cycles |
| | | | | $20 \times 20$ | 10,159 LUTs, 10,255 FFs, 60 DSPs, 22 BRAM18K, 330 MHz, 66,063 clock cycles |
| | | | | $16 \times 16$ | 9,033 LUTs, 10,097 FFs, 60 DSPs, 22 BRAM18K, 313 MHz, 45,861 clock cycles |
| | | | | $8 \times 8$ | 8,889 LUTs, 9,974 FFs, 60 DSPs, 22 BRAM18K, 330 MHz, 10,765 clock cycles |

posed, along with their *MATLAB* implementation, have shown how increasing precision in the presence of ill-conditioning might affect the obtained results in a negative way (by increasing error). Hence, in the face of an inadequate handling of precision (i.e., neither carefully selecting precision, nor applying a regularization or preconditioning method), the solution to a problem with perturbed data can contain large errors, especially if no attention is paid to the precision of the data used in the calculation.

From the analysis of trade-offs between truncation and precision, we observe larger errors when $\varepsilon$ is truncated to too few digits (Case Study 1). Conversely, errors are smaller when $\varepsilon$ is truncated to an adequate number of digits (Case Study 2).

Case studies 1 and 2 lead us to conclude that, if we aim at reducing error below a certain acceptable threshold, we need to determine an optimal precision point, which will prevent the system/model from becoming too sensitive to measurement errors.

On the other hand, the inverse $2 \times 2$ matrix calculation results with different FPGAs illustrate different trade-offs between hardware resources (use of LUTs, BRAMs, FFs, DSPs, critical paths, and IOBs), efficiency, and energy consumption as a consequence of increasing computational precision. The minimum clock period is obtained by implementing the architecture on a given FPGA device and the maximum clock frequency is the inverse of that period. Implementation results on the same devices are obtained in order to provide a fair comparison among different designs. These results describe the advantages and disadvantages of the *Spartan-7*, *Artix-7*, *Kintex-7*, and *Virtex-7* devices, and they show the benefits of the proposed architecture using state-of-the-art FPGAs in their different versions. These results show how most costs (use of LUTs, BRAMs, FFs, DSPs, critical paths, IOBs, efficiency, and energy consumption) more than double when double precision is used. In general, if more hardware resources are necessary, the critical paths can be larger. Additionally, the fact that not all devices were able to handle all precision variations shows that increasing precision will not be feasible in every scenario.

Specifically, we can see that *Virtex-7* and *Kintex-7* provide higher throughput than *Artix-7* and *Spartan-7*. The key element is full unrolling because it enables a design to increase

**Table 11. Advantages and disadvantages of low and high precision.**

| Advantages | Disadvantages |
|---|---|
| Low precision | |
| Fast computation. Low computation time. Low power consumption. Low space. | Sometimes, results will be imprecise (such as is the case of ill-conditioning). |
| High precision | |
| More precise results (except in 'problematic' situations, such as is the case of ill-conditioning). | Long computation time. High power consumption. High space. More complex solutions. Greater sensitivity to sensing errors. Regularization required, as in the case of ill-conditioning, or for determining optimal precision. |

throughput, although efficiency decreases. *Spartan-7* and *Artix-7* provide lower efficiency and lower throughput while using a similar amount of hardware resources. Table 11 summarizes the main advantages and disadvantages of using lower or higher precision levels.

The results presented here are as one would expect, except changes in the amount of resources used in calculations, which seem to grow not linearly with respect to changes in precision (although proving this is not in the scope of this work). In other words, more accurate results will be obtained when precision is higher, while calculations will require more time, energy, and hardware space. Nonetheless, one should note that precision should be carefully selected and/or regularization or preconditioning methods should be applied if non-ideal conditions are present, as in the case of ill-conditioned matrices. As a result, the model/system will be less sensitive to small errors, and considerable energy, time, and space would be saved.

As part of future work, we intend to optimize the fully unrolled architecture while analyzing better alternatives for improving the architecture presented in this work. The use of regularization methods alongside changes in precision will be explored in future work.

## Supporting information

**S1 File.**
(PDF)

**S1 Data.**
(XLSX)

## Acknowledgments

We would like to thank Editage (www.editage.com) for English language editing.

## Author Contributions

**Conceptualization:** Ignacio Algredo-Badillo, José Julio Conde-Mones, Carlos Arturo Hernández-Gracidas, María Monserrat Morín-Castillo, José Jacobo Oliveros-Oliveros.

**Data curation:** Ignacio Algredo-Badillo, José Julio Conde-Mones.

**Formal analysis:** Ignacio Algredo-Badillo, José Julio Conde-Mones, María Monserrat Morín-Castillo, José Jacobo Oliveros-Oliveros.

**Investigation:** Ignacio Algredo-Badillo, José Julio Conde-Mones, Carlos Arturo Hernández-Gracidas, María Monserrat Morín-Castillo, José Jacobo Oliveros-Oliveros.

**Methodology:** Ignacio Algredo-Badillo, José Julio Conde-Mones, Carlos Arturo Hernández-Gracidas, María Monserrat Morín-Castillo, José Jacobo Oliveros-Oliveros.

**Resources:** Ignacio Algredo-Badillo.

**Software:** Ignacio Algredo-Badillo, José Julio Conde-Mones.

**Validation:** Ignacio Algredo-Badillo, José Julio Conde-Mones, José Jacobo Oliveros-Oliveros.

**Visualization:** Ignacio Algredo-Badillo, José Julio Conde-Mones, Carlos Arturo Hernández-Gracidas.

**Writing – original draft:** Ignacio Algredo-Badillo, José Julio Conde-Mones, Carlos Arturo Hernández-Gracidas, María Monserrat Morín-Castillo, José Jacobo Oliveros-Oliveros.

**Writing – review & editing:** Ignacio Algredo-Badillo, José Julio Conde-Mones, Carlos Arturo Hernández-Gracidas, María Monserrat Morín-Castillo, José Jacobo Oliveros-Oliveros, Claudia Feregrino-Uribe.

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
