## [Decision Letter · Decision Letter 0]

6 Apr 2020

PONE-D-20-02712

An FPGA-based analysis of trade-offs in the presence of ill-conditioning and different precision levels in computations

PLOS ONE

Dear Dr. Hernández-Gracidas,

Thank you for submitting your manuscript to PLOS ONE. After careful consideration, we feel that it has merit but does not fully meet PLOS ONE’s publication criteria as it currently stands. Therefore, we invite you to submit a revised version of the manuscript that addresses the points raised during the review process.

We recommend that it should be revised taking into account the changes requested by the reviewers. To speed the review process, the revised manuscript will only be reviewed by the Academic Editor in the next round.

We would appreciate receiving your revised manuscript by May 21 2020 11:59PM. To enhance the reproducibility of your results, we recommend that if applicable you deposit your laboratory protocols in protocols.io, where a protocol can be assigned its own identifier (DOI) such that it can be cited independently in the future. For instructions see: http://journals.plos.org/plosone/s/submission-guidelines#loc-laboratory-protocols

We look forward to receiving your revised manuscript.

Kind regards,

Baogui Xin, Ph.D.

Academic Editor

PLOS ONE

Journal Requirements:

Reviewers' comments:

Reviewer's Responses to Questions

**Comments to the Author**

1. Is the manuscript technically sound, and do the data support the conclusions?

Reviewer #1: Yes

Reviewer #2: Partly

2. Has the statistical analysis been performed appropriately and rigorously? 

Reviewer #1: Yes

Reviewer #2: N/A

3. Have the authors made all data underlying the findings in their manuscript fully available?

Reviewer #1: Yes

Reviewer #2: Yes

4. Is the manuscript presented in an intelligible fashion and written in standard English?

Reviewer #1: Yes

Reviewer #2: No

5. Review Comments to the Author

Reviewer #1: This manuscript needs minor revision to describe acronyms and present the results in a Table that summarizes main aspects of hardware resources and performances. This and other issues can be impored as follows:

In line 167 you claim that..

In traditional digital systems, such as FPGAs and microcomputers, the 167

simplest numeric representation corresponds to fixed-point notation, where p bits can 168

represent up to 2p different consecutive integer numbers.

This is not true, fixed-point notation is not traditional. Fixed-point notation is much more simple to implemented on an FPGA for some applications like the one given in the paper: ...FPGA-based implementation of chaotic oscillators by applying the numerical method based on trigonometric polynomials, AIP Advances 8 (7), 075217, 2018

As you netion in line 175:

Floating-point notation, on the other hand, is used to represent rational 175

numbers, with a mantissa (M ) and an exponent (E ) (in a similar fashion to scientific 176

From these paragraphs it is straightforward that floating-point is more exact and allows representations of big numbers than by using fixed-point notation. You need to highlight this in your manuscript.

You need to describe all acronyms, and some can be given before you use them, see for example that from line 183 you can add the acronyms as follows:

Popular precisions used for floating point are: i) half precision (HP), ii) single 183

Precision (SP), iii) double precision (DP), iv) extra precision (EP), and v) quadruple precision (see Table 1184

Other acronyms are in line 193, and need definition...

IoT, Domotics, IIoT, and 193

In Section Discussion you give some details of hardware resources in the text, but it can be more attractive for the reader if you prepare a Table listing the used resources (see an exmaple in the article mentioned above) in the different FPGAs for the different lengths of bits and also including the precision and exactness for the cases of study.

Reviewer #2: This revised version provides supported data and revised points according to the revised comments.

The data of hardware costs and precision seem to be sufficient.

But the main problem of this version is the motivation and case of ill-condition.

I recommend that realistic ill-condition and its analysis can be the key of this paper.

The ill-condition of this paper adopts truncation. But this truncation for producing ill-condition is not explained. Is there any other ill conditions except for the truncation?

Matrix vector multiplication is important in many application. But this paper adopt just 2x2 multiplication, which does not show any scalibiltity. What happens in nxn multiplication?

In introduction, many expressions are not related to ill-condition.

Overall, several expressions should be rewritten, and scalability of ill-condition should be explained.

In rebuttal, the original comments can be attached.

6. PLOS authors have the option to publish the peer review history of their article (what does this mean?). If published, this will include your full peer review and any attached files.

Reviewer #1: No

Reviewer #2: No

---

## [Author Response · Author response to Decision Letter 0]

21 May 2020

We appreciate enormously the opportunity of submitting a revised manuscript as well as the comments made by the Reviewers and the Editor. This feedback has been extremely helpful for detecting flaws in the document. Hopefully, in this revised version, we have been able to improve its quality enough to meet the quality standards of PLOS ONE, thanks to your help.

To facilitate tracking the changes mentioned in this letter, we use the following format in the file labeled ‘Revised Manuscript with Track Changes’:

RESPONSES TO REVIEWER #1

Reviewer # 1: This manuscript needs minor revision to describe acronyms and present the results in a Table that summarizes main aspects of hardware resources and performances. This and other issues can be impored as follows:

Comment 1.

In line 167 you claim that.. In traditional digital systems, such as FPGAs and microcomputers, the 167 simplest numeric representation corresponds to fixed-point notation, where p bits can 168 represent up to 2p different consecutive integer numbers.

This is not true, fixed-point notation is not traditional. Fixed-point notation is much more simple to implemented on an FPGA for some applications like the one given in the paper: ...FPGA-based implementation of chaotic oscillators by applying the numerical method based on trigonometric polynomials, AIP Advances 8 (7), 075217, 2018

As you netion in line 175: Floating-point notation, on the other hand, is used to represent rational 175 numbers, with a mantissa (M ) and an exponent (E ) (in a similar fashion to scientific 176

From these paragraphs it is straightforward that floating-point is more exact and allows representations of big numbers than by using fixed-point notation. You need to highlight this in your manuscript.

Answer: We recognize that the reviewer is right and apologize for the erroneous claim. To respond to the observations made by Reviewer #1, we have implemented the following changes in the revised manuscript to rectify our previously inaccurate claim and emphasize the characteristics of fixed-point notation:

In the Precision section, regarding the claim of fixed-point notation being traditional,

 Previously, lines 200–201 stated the following: In traditional digital systems, such as FPGAs and microcomputers, the simplest numeric representation... 

This has been updated to mention (lines 207–208) the following: In several applications and digital systems, such as FPGAs and microcomputers, an important numeric representation... 

In the Precision section, the following paragraph was added to complement the existing content originally in lines 208–209 and currently in lines 215–216, emphasizing the characteristics of fixed-point notation:

Added paragraph (lines 226–230): When comparing fixed-point and floating-point representations, the latter is more exact and allows the representation of bigger and smaller numbers than the former. With the drawback that its software and hardware implementations are more complex, which necessitates the analysis and development of new methods to improve the performance while using floating-point representations.

Comment 2.

You need to describe all acronyms, and some can be given before you use them, see for example that from line 183 you can add the acronyms as follows:

Popular precisions used for floating point are: i) half precision (HP), ii) single 183 Precision (SP), iii) double precision (DP), iv) extra precision (EP), and v) quadruple precision (see Table 1184

Other acronyms are in line 193, and need definition... IoT, Domotics, IIoT, and 193

Answer: We apologize for our oversight regarding the correct use of acronyms. In response to this observation, we introduce each acronym before its first use as follows:

In the Precision section,

Previously, lines 216–217 stated the following: Popular precision levels used for floating point calculations are: i) half precision, ii) single precision, iii) double precision, iv) extra precision, and v) quadruple precision (see Table 1).

These have been updated to the following (lines 223–225): Popular precision levels used for floating point calculations are i) half precision (HP), ii) single precision (SP), iii) double precision (DP), iv) extra precision (EP), and v) quadruple precision (QP). Further details are provided in Table 1.

In the Methods section,

Previously, lines 226–227 stated the following: ... for IoT, domotics, IIoT, and Industry 4.0 technologies; ...

These have been updated as follows (lines 238–239): ... for Internet of Things (IoT), domotics, Industrial IoT (IIoT), and Industry 4.0 technologies; ...

In the Methods section,

Line 239 previously stated the following: ... such as Gauss-Jordan elimination and QR decomposition.

Now, it states the following (lines 251–253): ... such as Gauss–Jordan elimination and QR decomposition (where a matrix M is decomposed into a product M=QR of an orthogonal matrix Q and an upper triangular matrix R).

 In the Methods section,

Lines 284–285 previously stated the following: In this work, four designs using the same architecture were evaluated. Each design (HP, SP, DP, and EP) is ...

This was reworded as follows (lines 297–298): In this work, four designs using the same architecture were evaluated: HP, SP, DP, and EP. Each design is ... 

In the Results section,

Previously, lines 409–411 stated the following: ... such as the number of slices, LUTs, flip-flops, BRAMs, digital signal processors (DSPs), and IOBs, along with the minimum clock period and power consumption.

Now, the updated text is as follows (lines 428–431): ... such as the number of slices, look-up tables (LUTs), flip-flops (FFs), block random access memories (BRAMs), digital signal processors (DSPs), and IOBs, along with the minimum clock period and power consumption.

In the Results section,

Line 423 previously stated as follows: ..., LUTRAM, FFs ...

Now, it states the following (lines 447–448): ..., look-up table random access memory (LUTRAM), FFs ...

In the Comparisons section,

Previously, lines 570–571 stated as follows: ... using VHDL and ...

Now, it states the following (lines 602–603): ... using very high speed integrated circuit hardware description language (VHDL) and ...

Thus, all acronyms have been described at their first occurrences.

Comment 3.

In Section Discussion you give some details of hardware resources in the text, but it can be more attractive for the reader if you prepare a Table listing the used resources (see an exmaple in the article mentioned above) in the different FPGAs for the different lengths of bits and also including the precision and exactness for the cases of study.

Answer: The results are originally presented using plots with the purpose of making them easier to read for the general scientific community. However, we agree that using tables can be beneficial for certain readers. Consequently, a paragraph and four tables have been introduced, in addition to the plotted results. The tables were prepared based on the format used in the example article, as suggested by the Reviewer.

Added paragraph (lines 441–444): Considering these metrics, the hardware architecture was implemented using four technologies (Artix-7, Kintex-7, Spartan-7, and Virtex-7) for the MP and SP versions (see Tables 5 and 6, respectively), and one technology (Virtex-7) for the DP and EP versions (see Tables 7 and 8, respectively). 

RESPONSES TO REVIEWER #2

Reviewer # 2: This revised version provides supported data and revised points according to the revised comments.

Comment 1.

The data of hardware costs and precision seem to be sufficient. But the main problem of this version is the motivation and case of ill-condition.

I recommend that realistic ill-condition and its analysis can be the key of this paper.

Answer: In response to the Reviewer’s recommendation, we expanded the explanation that motivates this work, highlighting the ill-posedness (in the sense of Hadamard) of the operational equations that appear when practical problems are modeled using integral or differential equations. In particular, we emphasize that the ill-posedness of these problems leads to systems of algebraic equations with ill-conditioned matrices. Furthermore, we illustrate the generated effect when the inverse of a matrix is calculated when its elements are truncated. This is mentioned in various parts of the document, as detailed below.

In the Abstract,

Previously, it stated: We first demonstrate some examples of real-life problems where ill-conditioning is present in matrices obtained from discretization of the operational equations that model these problems. If these matrices are not handled properly (i.e., if ill-conditioning is not taken into account), we can obtain large errors in the calculation of their inverses.

The updated text reads as follows: We first demonstrate some examples of real-life problems where ill-conditioning is present in matrices obtained from the discretization of the operational equations (ill-posed in the sense of Hadamard) that model these problems. If these matrices are not handled appropriately (i.e., if ill-conditioning is not considered), large errors can result in the computed solutions to the systems of equations in the presence of errors. Furthermore, we illustrate the generated effect in the calculation of the inverse of an ill-conditioned matrix when its elements are approximated by truncation.

In the Introduction section,

Lines 21–24 stated: whereas some systems of algebraic equations have matrices that can be obtained when some linear operational equations in infinite dimensions are discretized. These operational equations can be derived when a problem is modeled through integral or differential equations.

This has been updated to the following (lines 21–25): whereas some systems of algebraic equations have (ill-conditioned) matrices that can be obtained when some linear operational equations in infinite dimensions are discretized. These operational equations can be ill-posed in the sense of Hadamard [2], which leads to ill-conditioned matrices, and can be derived when a problem is modeled by integral or differential equations.

In the Introduction section,

Previously, lines 47–49 read: This operational equation can be discretized, which leads to a linear system of equations with an ill-conditioned matrix that can produce numerical instability.

Now, it states (lines 48–56) the following: This operational equation, which is ill-posed in the sense of Hadamard [2], can be discretized, leading to a linear system of equations with an ill-conditioned matrix that can result in numerical instability in the presence of errors. Notably, the operator K is linear, compact, and injective, and φ and V belong to appropriate Hilbert spaces of infinite dimension, which imply that K^-1 (the inverse operator of K) is not continuous [2]. Therefore, the matrices obtained when the operational equation Kφ=V is discretized are ill-conditioned. These operational equations appear in various practical problems, such as the source and potential identification problems.

As a consequence of point 3 above, the matrices obtained by the discretization of ill-posed operational equations are ill-conditioned. Thus, the systems of algebraic equations associated with these matrices exhibit numerical instability in the presence of errors. This is inherent in the characteristics of the operational equations that are used to study real-life problems, such as source and parameter identification inverse problems. If the numerical instability (due to ill-conditioning) is overlooked, the solution that is obtained to the system of algebraic equations with errors can be far from the solution to the same system without errors. This situation is illustrated in Table 3. Therefore, we conclude that it is essential to appropriately handle ill-conditioning (which is one of the reasons for numerical instability). As aforementioned, this appears in practical problems, such as the inverse source and potential identification problems associated with electroencephalography and electrocardiography [3,5–7,10–12].

Comment 2.

The ill-condition of this paper adopts truncation. But this truncation for producing ill-condition is not explained. Is there any other ill conditions except for the truncation?

Answer: We apologize for the ambiguity. We have now emphasized that the ill-conditioning of a matrix does not depend on errors (e.g., by rounding up and down, which includes truncation) since it is inherent in the matrix itself. We have included the following remark in the manuscript to clarify this:

In the Results section,

Added a paragraph (lines 395–400): Remark: The ill-conditioning of a matrix does not depend on errors (e.g., by rounding up and down, which includes truncation) since it is inherent in the matrix itself. Ill-conditioning amplifies any errors that appear in the system, such as rounding errors or errors inherent in the measurement devices. Conversely, truncation is occasionally used for saving hardware or software resources, or both. Hence, we analyzed the effects of truncation with ill-conditioned matrices.

We hope this Remark, along with the following fragment already in the previous version of the manuscript, fully clarifies this explanation (originally lines 377–381, now lines 390–394):

As we can observe in this case study, and in general, ill-conditioning in a matrix can lead to a situation wherein the solution to a system with erroneous data can be far from the solution with exact data (as shown in Table 3). This fact is independent of the numerical method used to compute the solution to the system, since similar results are obtained for various methods.

Comment 3.

Matrix vector multiplication is important in many application. But this paper adopt just 2 x2 multiplication, which does not show any scalibiltity. What happens in nxn multiplication?

Answer: We agree with your observation that scalability is a highly relevant aspect. Hence, we have included the following statements to clarify our approach and emphasize how scalability can be achieved with the architecture proposed in this paper:

In the Results section,

Added a paragraph (lines 522–529): The proposed architecture has a parallel and specific structure, which is scalable because it is based on invertible matrices using the adjoint method. If new mathematical models require other matrix sizes (in general a size of n×n), more modules are required according to the operation count, for example, [n!+n^2+3((n(n+1))/2 − 3)] multipliers, and [n!/2+((n(n+1))/2 -3)] adders, where n!/2 determinants of size 2×2 are calculated. However, there are other hardware-design techniques that are more suitable for reducing modules, which will be explored in future works.

Comment 4.

In introduction, many expressions are not related to ill-condition.

Answer: We apologize that the contents of the Introduction caused this perception. The example related to the Patriot missile failure, which is, indeed, not directly associated with ill-conditioning, was included to present this study in a more general context, i.e., the adequate handling of errors for algorithms.

With respect to the operational equation Kφ=V, we emphasize its relationship with the ill-conditioning of a matrix, by expanding the corresponding paragraph.

In the Introduction section,

Previously, lines 47–49 stated: This operational equation can be discretized, which leads to a linear system of equations with an ill-conditioned matrix that can produce numerical instability.

Now, lines 48–56 state: This operational equation, which is ill-posed in the sense of Hadamard [2], can be discretized, leading to a linear system of equations with an ill-conditioned matrix that can result in numerical instability in the presence of errors. Notably, operator K is linear, compact, and injective, and φ and V belong to appropriate Hilbert spaces of infinite dimension, which imply that K^-1 (the inverse operator of K) is not continuous [2]. Therefore, the matrices obtained when the operational equation Kφ=V is discretized are ill-conditioned. These operational equations appear in various practical problems, such as the source and potential identification problems.

Comment 5.

Overall, several expressions should be rewritten, and scalability of ill-condition should be explained.

Answer: We have performed several reviews and proof-reading rounds on the manuscript to detect expressions that needed to be re-written. Additionally, both the original manuscript and this letter, containing the revisions made in response to the reviewers' comments, have been edited using the professional service by Editage. We hope that the professional editing, along with the aforementioned changes, sufficiently address the issue raised in this comment.

---

## [Editor Report · Decision Letter 1]

26 May 2020

An FPGA-based analysis of trade-offs in the presence of ill-conditioning and different precision levels in computations

PONE-D-20-02712R1

Dear Dr. Hernández-Gracidas,

We are pleased to inform you that your manuscript has been judged scientifically suitable for publication and will be formally accepted for publication once it complies with all outstanding technical requirements.

With kind regards,

Baogui Xin, Ph.D.

Academic Editor

PLOS ONE
---

## [Editor Report · Acceptance letter]

5 Jun 2020

PONE-D-20-02712R1 

An FPGA-based analysis of trade-offs in the presence of ill-conditioning and different precision levels in computations 

Dear Dr. Hernández-Gracidas:

I'm pleased to inform you that your manuscript has been deemed suitable for publication in PLOS ONE. Congratulations! Your manuscript is now with our production department. 

Kind regards, 

on behalf of

Professor Baogui Xin 

Academic Editor

PLOS ONE